# C²GSPG: Confidence-calibrated Group Sequence Policy Gradient towards Self-aware Reasoning

## Abstract

Reinforcement Learning (RL) methods, exemplified by Group Relative Policy Optimization (GRPO) and its variants, play a central role in developing reasoning models. However, these methods often suffer from a critical overconfidence issue, which prevents them from achieving self-aware reasoning models. In this study, we propose a simple yet effective confidence-calibration group sequence policy gradient method, called C²GSPG, which simultaneously enhances reasoning performance while suppressing overconfidence. In principle, we propose a Group Sequence Policy Gradient (GSPG) framework for learning reasoning models, which eliminates the token-level bias commonly appearing in GRPO and its variants. In this framework, we define the model confidence for each reasoning problem using the normalized sequence-level probability, and then apply a cross-entropy regularizer to calibrate the model confidence to the sequence's reward. We demonstrate that the confidence calibration regularizer and GSPG are collaborative for binary rewards, as their objectives always share the same gradient direction. For non-binary rewards, we apply nonlinear reward normalization and adaptive regularizer clipping, mitigating the potential conflict between the two objectives. Applying C²GSPG to post-train large language models in logical and mathematical reasoning tasks, we show its superiority over state-of-the-art methods in both reasoning accuracy and confidence calibration.

## 1 Introduction

Reinforcement Learning (RL) has become a pivotal post-training technique for enhancing the advanced reasoning capabilities of Large Language Models (LLMs). Unlike traditional Supervised Fine-Tuning (SFT), RL-based methods utilize verifiable rewards to enable models to explore and refine their reasoning pathways autonomously. Among prominent RL algorithms, the recent Group Relative Policy Optimization (GRPO) (Shao et al., 2024) has garnered significant attention. Unlike PPO (Schulman et al., 2017) and DPO (Rafailov et al., 2023), GRPO eliminates the need for a value model, thereby simplifying model design and improving computational efficiency. Along this direction, many variants of GRPO have been proposed, e.g., DAPO (Yu et al., 2025), GPG (Chu et al., 2025), and GSPO (Zheng et al., 2025), and demonstrated superior performance in logical and mathematical reasoning tasks.

While GRPO and its variants can improve the reasoning ability of LLMs, they often induce an *overconfidence* issue — the post-trained models cannot be aware of their mistakes (Bereket & Leskovec, 2025; Chhikara, 2025). In the paradigm of RL, this issue is reflected by the mismatch between the probabilities of generated sequences and the rewards associated with the sequences, i.e., the high-probability sequences may lead to low rewards, as shown in Figure 1(a). To mitigate the overconfidence issue, some methods first modulate reward functions based on empirical or learnable model confidences, and then reduce the significance of high-confidence but low-quality samples during training (Zhou et al., 2025; Damani et al., 2025; Wen et al., 2024; Zeng et al., 2025). However, these methods often suffer from degraded reasoning accuracy or insufficient confidence calibration because they ignore the potential conflicts between policy optimization and confidence calibration. How to pursue confidence calibration without compromising policy optimization, especially in scenarios with non-binary rewards, remains an open problem.

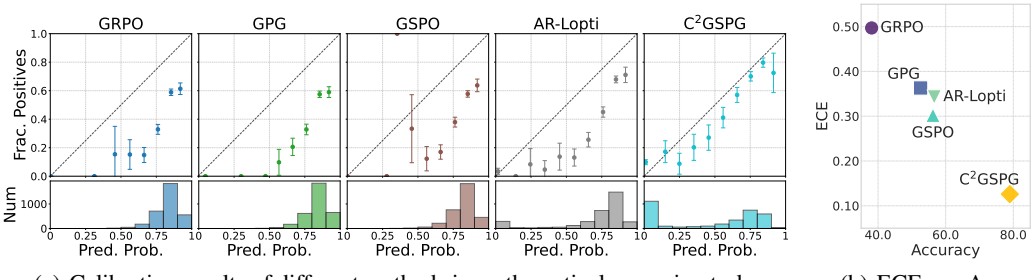

(a) Calibration results of different methods in mathematical reasoning tasks

(b) ECE v.s. Acc

Figure 1: The comparison for various methods on the consistency between model confidences and rewards. The panel (a) presents reliability diagrams of various methods on six mathematical reasoning tasks, demonstrating our method's effective calibration. The panel (b) shows Expected Calibration Error (ECE) against Accuracy on the "Knights and Knaves" logic puzzle (K&K) dataset (Xie et al., 2024), where our method reaches the best performance quadrant (high accuracy and low ECE).

In this study, we propose a simple yet effective RL method, called Confidence-calibrated Group Sequence Policy Gradient ($C^2$GSPG), which yields robust reasoning models with calibrated confidence. For each reasoning problem, our method generates a group of sequences and defines model confidences of the sequences using normalized sequence-level probabilities. Based on the confidence, we propose a confidence-modulated advantage function and derive a Group Sequence Policy Gradient (GSPG) framework for training reasoning models. Similar to existing sequence-based RL methods (Zheng et al., 2025), GSPG eliminates the token-level bias of GRPO during training (Yang et al., 2025b). Furthermore, we introduce a Binary Cross-Entropy (BCE) regularizer into the framework to encourage the consistency between each sequence's model confidence and the corresponding reward, resulting in the proposed $C^2$GSPG method. We theoretically demonstrate a collaborative relationship between policy optimization and our regularizer in binary-reward scenarios — these two objectives share the same gradient direction in all situations. This insight motivates us to propose nonlinear reward normalization and adaptive regularizer clipping to mitigate the potential conflict between the two objectives in the more complex, non-binary reward cases.

We compare $C^2$GSPG with GRPO and its variants on logical and mathematical reasoning tasks. The results demonstrate that our method effectively mitigates the model overconfidence problem while also improving reasoning ability. Moreover, Figure 1(a) validates that, unlike baselines which often exhibit poorly separated distributions, $C^2$GSPG successfully cultivates a bimodal confidence profile: high confidence is predominantly assigned to correct answers, while incorrect answers are given low confidence. Furthermore, Figure 1(b) illustrates that $C^2$GSPG consistently achieves a higher accuracy on the validation set while simultaneously driving down the Expected Calibration Error (ECE) more effectively than competing approaches.

## 2 PRELIMINARIES AND RELATED WORK

### 2.1 GROUP RELATIVE POLICY OPTIMIZATION AND ITS VARIANTS

RL-based fine-tuning has become a cornerstone for improving the reasoning ability of LLMs (Yang et al., 2025a; Shao et al., 2024; Comanici et al., 2025; Jaech et al., 2024). In the context of RL, an LLM is often represented as a parametric policy, denoted as $\pi_\theta$. Given the current policy, denoted as $\pi_{\theta_{\text{old}}}$, **GRPO** (Shao et al., 2024) samples a group of $G$ responses, denoted as $\{o_i\}_{i=1}^G$, for each prompt $q$ according to the distribution $\pi_{\theta_{\text{old}}}(\cdot|q)$. Each response is a sequence of tokens (i.e., $o_i = \{o_{i,t}\}_{t=1}$, and $o_{i,t}$ is the $t$-th token of $o_i$), which includes a step-by-step reasoning chain leading to a final answer. A scalar reward $r_i$ is assigned to each response $o_i$. Instead of applying the reward directly, GRPO computes a relative advantage for each response by comparing it to other responses sampled for the same prompt, i.e.,

$$\hat{A}_i = \frac{r_i - m}{\sigma}, \text{ where } m = \frac{1}{G}\sum_{i=1}^G r_i \text{ and } \sigma = \sqrt{\frac{1}{G}\sum_{i=1}^G (r_i - m)^2}. \tag{1}$$

GRPO incorporates this relative advantage into a clipped surrogate objective, which constrains the magnitude of policy updates. The overall objective function to be maximized is:

$$\mathcal{J}_{\text{GRPO}}(\theta) = \mathbb{E}_{\{o_i\} \sim \pi_{\theta_{\text{old}}}} \frac{1}{G} \sum_{i=1}^{G} \frac{1}{|o_i|} \sum_{t=1}^{|o_i|} \min\left[ \frac{\pi_{\theta,i,t}}{\pi_{\theta_{\text{old}},i,t}} \hat{A}_i, \text{clip}_\epsilon\left( \frac{\pi_{\theta,i,t}}{\pi_{\theta_{\text{old}},i,t}} \right) \hat{A}_i \right] - \gamma D_{\text{KL}}(\pi_\theta \| \pi_{\text{ref}}). \quad (2)$$

Here, we denote $\pi_\theta(o_{i,t}|o_{i,<t}, q)$ as $\pi_{\theta,i,t}$ for short. $|o_i|$ represents the length of $o_i$. The function $\text{clip}_\epsilon(\cdot) = \min(\max(\cdot, 1-\epsilon), 1+\epsilon)$, and $\pi_{\text{ref}}$ denotes a reference policy (e.g., the pre-trained LLM).

Recently, some efforts have been made to improve GRPO (Chu et al., 2025; Zheng et al., 2025; Yu et al., 2025; Yang et al., 2025b; Li et al., 2025). Yang et al. (2025b) uses Advantage Reweighting and Low-Probability Token Isolation (**AR-Lopti**) to mitigate the over-dominance of low-probability tokens during training. Group Policy Gradient (**GPG**) (Chu et al., 2025) updates the policy by maximizing the logarithmic token probabilities weighted by unnormalized advantages, i.e.,

$$\mathcal{J}_{\text{GPG}}(\theta) = \mathbb{E}_{\{o_i\} \sim \pi_{\theta_{\text{old}}}} \frac{1}{\sum_{j=1}^{G} |o_j|} \sum_{i=1}^{G} \sum_{t=1}^{|o_i|} A_i \cdot \log \pi_{\theta,i,t}, \quad \text{where } A_i = r_i - m. \quad (3)$$

Group Sequence Policy Optimization (**GSPO**) (Zheng et al., 2025) replaces the token-level probability ratio $\pi_{\theta,i,t}$ with a sequence-level one, denoted as $s_{\theta,i}$. Its objective then incorporates this ratio into a clipped surrogate function, closely resembling the PPO structure:

$$\mathcal{J}_{\text{GSPO}}(\theta) = \mathbb{E}_{\{o_i\} \sim \pi_{\theta_{\text{old}}}} \frac{1}{G} \sum_{i=1}^{G} \min\left[ s_{\theta,i} \hat{A}_i, \text{clip}_\epsilon(s_{\theta,i}) \hat{A}_i \right], \quad \text{where } s_{\theta,i} = \left( \frac{\pi_\theta(o_i|q)}{\pi_{\theta_{\text{old}}}(o_i|q)} \right)^{\frac{1}{|o_i|}}. \quad (4)$$

The sequence-level ratio $s_{\theta,i}$ is defined as the geometric mean of the token-level probability ratios, which is implemented by $\exp\left( \frac{1}{|o_i|} \sum_{t=1}^{|o_i|} \log \frac{\pi_{\theta,i,t}}{\pi_{\theta_{\text{old}},i,t}} \right)$ in practice.

## 2.2 THE OVERCONFIDENCE ISSUE AND CURRENT SOLUTIONS

As aforementioned, GRPO and its variants often result in overconfident models (Leng et al., 2025; Bereket & Leskovec, 2025), which necessitates confidence calibration to ensure their reliability. The practical risks posed by such overconfidence have been well-studied (Melotti et al., 2022; Omar et al., 2024; Griot et al., 2025). In risk-sensitive applications such as disease diagnosis and auto-driving, confidence scores are key signals for introducing human auditing or fallback mechanisms. Overconfident models are unreliable (and inapplicable) in such applications because their mistakes are hard to predict. An LLM's confidence is essentially its own prediction of how likely its answer is to be right, while calibration is the test of how reliable the prediction is — it checks whether the model's stated confidence levels match its true success rate (Yoon et al., 2025b). Several methods have been developed to estimate this confidence. Gupta et al. (2024) uses the token-level probability of the answer to form a confidence score. Kadavath et al. (2022) employs other models to predict correctness. Xu et al. (2024) leverages labeled confidence scores datasets to fine-tune models. The methods in (Yoon et al., 2025b; Damani et al., 2025) elicit verbalized confidence, which prompts the model to articulate its own confidence estimate directly within its output.

Traditional confidence calibration approaches often operate at inference time by prompting the model to self-assess its understanding (Wen et al., 2024) or by introducing distractors (Chhikara, 2025). However, they often focus on scenarios of verbalized confidence, with unverified applicability to token-probability-based confidence. Recent works have sought to integrate calibration directly through regularization. For example, RLCR (Damani et al., 2025) designs a hybrid reward combining accuracy with a Brier calibration score. Taming LLMs (Leng et al., 2025) dynamically modifies reward values during PPO training based on recent performance. Similarly, CoDaPO (Zhou et al., 2025) modulates the advantage function with confidence and difficulty weights. Despite their contributions, these training-based methods often struggle to achieve a trade-off between confidence and accuracy, sometimes sacrificing performance or yielding suboptimal calibration.

# 3 CONFIDENCE-CALIBRATED GROUP SEQUENCE POLICY GRADIENT

To mitigate the model overconfidence issue while maintaining high reasoning performance, we propose a method, called Confidence-Calibration Group Sequence Policy Gradient (**C²GSPG**), that

introduces a novel objective function to explicitly align a model's self-assessed confidence with its task performance. **Our method is motivated by a desideratum commonly used in confidence calibration (Grabinski et al., 2022; Wen et al., 2024; Yoon et al., 2025a): the probability of generated sequence should reflect the model confidence corresponding to the prompt/question.** Therefore, our method defines model confidence directly from the model's sequence-level probabilities, rather than using verbalized confidence statements (Bereket & Leskovec, 2025). This is universally applicable and provides a more direct measure of the model's internal certainty (Gupta et al., 2024). Formally, for the $i$-th sequence $o_i$ corresponding to the prompt $q$, we compute the model confidence of $o_i$ as

$$c_{\theta,i} := \pi_\theta(o_i|q)^{\frac{1}{|o_i|}} = \exp\Big(\frac{1}{|o_i|}\sum\nolimits_{t=1}^{|o_i|} \log \pi_{\theta,i,t}\Big). \qquad (5)$$

Here, $\pi_\theta(o_i|q)^{\frac{1}{|o_i|}}$ denotes normalized sequence-level probability, which can be computed based on the token-level probabilities $\{\pi_{\theta,i,t}\}$. The normalization eliminates the impact of sequence length.

### 3.1 IMPLEMENTATION AND ANALYSIS IN BINARY REWARD SCENARIOS

In principle, the objective function of our method consists of a policy optimization term and a confidence calibration regularizer, both of which hinge on the confidence $c_{\theta,i}$. In the following content, we will introduce its implementation and analyze its rationality in theory. For simplification, we first implement and analyze our method in binary reward scenarios. For each prompt $q$ and a group of generated sequences $\{o_i\}_{i=1}^G$, we apply a group sequence policy gradient loss with a binary cross-entropy regularizer to implement this objective, i.e.,

$$\mathcal{J}_{\mathrm{C^2GSPG}}(\theta) = \mathbb{E}_{\{o_i\}\sim\pi_{\theta_{\mathrm{old}}}} \frac{1}{G}\sum_{i=1}^G \Big[\underbrace{\tilde{A}_i \cdot \log c_{\theta,i}}_{\text{Policy gradient}} + \beta\underbrace{[r_i \log c_{\theta,i} + (1-r_i)\log(1-c_{\theta,i})]}_{\text{Binary cross-entropy regularizer}}\Big], \qquad (6)$$

where $r_i \in \{0,1\}$ is binary reward achieved by $o_i$. $\tilde{A}_i$ is a normalized advantage defined as

$$\tilde{A}_i = \frac{r_i - m}{1 - c_{\theta_{\mathrm{old}},i}} = \frac{r_i - \frac{1}{G}\sum_{j=1}^G r_j}{1 - c_{\theta_{\mathrm{old}},i}}, \qquad (7)$$

where $c_{\theta_{\mathrm{old}},i}$ denotes the confidence of the old policy, and $m$ is the average of the binary rewards in the group. For each sequence $o_i$, the policy gradient term $\tilde{A}_i \cdot \log c_{\theta,i}$ in (6) pursues the consistency between the model confidence and the advantage function — the high-confident sequence should achieve a high advantage. The cross-entropy regularizer $\beta[r_i \log c_{\theta,i} + (1-r_i)\log(1-c_{\theta,i})]$ penalizes the discrepancy between the reward and the model confidence of the sequence, whose significance is controlled by the hyperparameter $\beta > 0$. This regularizer pushes the model confidence towards 1 for high-reward sequences and towards 0 for low-reward sequences.

**Remark.** The design of $\tilde{A}_i$ integrates group-level advantage with model confidence, thereby introducing confidence as a fine-grained modulation factor across sequences within the same group. When the answer is incorrect, we have $r_i - m \leq 0$, in which a smaller $c_i$ leads to a larger $\tilde{A}_i$. Conversely, when the answer is correct and $r_i - m \geq 0$, a larger $c_i$ results in a larger $\tilde{A}_i$. In other words, this modulation aligns with the objective of model calibration, encouraging the model to be more confident for truly advantageous sequences while reducing confidence for disadvantageous ones.

By defining our objective in this manner, we formally analyze its underlying optimization dynamics. In particular, according to the definition in (5), the gradient of $c_{\theta,i}$ at $\theta$ is

$$\nabla_\theta c_{\theta,i} = c_{\theta,i} \cdot \Big(\frac{1}{|o_i|}\sum\nolimits_{t=1}^{|o_i|} \nabla_\theta \log \pi_{\theta,i,t}\Big). \qquad (8)$$

When computing the gradient of $\mathcal{J}_{\mathrm{C^2GSPG}}(\theta)$ at $\theta$, we apply the gradient chain rule and obtain

$$\nabla_\theta \mathcal{J}_{\mathrm{C^2GSPG}}(\theta) = \mathbb{E}_{\{o_i\}\sim\pi_{\theta_{\mathrm{old}}}} \frac{1}{G}\sum_{i=1}^G \Big(\tilde{A}_i + \beta\frac{r_i - c_{\theta,i}}{1 - c_{\theta,i}}\Big)\Big(\frac{1}{|o_i|}\sum_{t=1}^{|o_i|}\nabla_\theta \log \pi_{\theta,i,t}\Big)$$

$$= \mathbb{E}_{\{o_i\}\sim\pi_{\theta_{\mathrm{old}}}} \frac{1}{G}\sum_{i=1}^G \Big(\frac{r_i - m}{1 - c_{\theta_{\mathrm{old}},i}} + \beta\frac{r_i - c_{\theta,i}}{1 - c_{\theta,i}}\Big)\Big(\frac{1}{|o_i|}\sum_{t=1}^{|o_i|}\nabla_\theta \log \pi_{\theta,i,t}\Big). \qquad (9)$$

As shown in (9), the rationality and effectiveness of our proposed training objective hinge on the collaboration between the policy optimization term and the confidence-calibration regularizer, whose directions are determined by $r_i - m$ and $r_i - c_{\theta,i}$, respectively. As shown in the following proposition, these two terms work in synergy for binary rewards — the model's confidence is adjusted in a manner that is consistent with the policy update.

Table 1: The directions of different terms in the gradient in (9).

| Weight | Correct $o_i$ | Incorrect $o_i$ |
|---|---|---|
| $r_i - m$ | $\geq 0$ | $\leq 0$ |
| $r_i - c_{\theta,i}$ | $\geq 0$ | $\leq 0$ |

**Proposition 3.1.** *Given binary rewards $r_i \in \{0,1\}$, the group mean reward $m \in (0,1)$ and the model's confidence $c_{\theta,i} \in (0,1)$, the policy optimization direction $r_i - m$ and the regularization direction $r_i - c_{\theta,i}$ are consistent, that is, they always share the same sign.*

The proof of this proposition is straightforward. As shown in Table 1, for a correct response ($r_i = 1$), the policy optimization direction $r_i - m$ and the regularization direction $r_i - c_{\theta,i}$ are both nonnegative, while for an incorrect response ($r_i = 0$), they are both non-positive. Such consistency ensures that the model becomes more confident in correct responses and less confident in incorrect ones.

To further highlight the effectiveness of C$^2$GSPG, given a sequence $o_i$, we compare it with several representative methods (including GRPO (Shao et al., 2024), GPG (Chu et al., 2025), AR-Lopti (Yang et al., 2025b), and GSPO (Zheng et al., 2025)) on their gradient formulations. Table 2 shows that C$^2$GSPG offers the following two advantages.[1]

Table 2: A comparison of the gradient per sequence. We denote $\nabla_\theta \log \pi_{\theta,i,t}$ as $g_{\theta,i,t}$ for short. The methods having red terms treat tokens of a sequence differently, while the others treat all tokens evenly. The blue terms show the impact of model confidence on advantage.

| Method | Gradient per sequence |
|---|---|
| GRPO | $\frac{r_i - m}{|o_i|\sigma} \sum_{t=1}^{|o_i|} \frac{\pi_{\theta,i,t}}{\pi_{\theta_{\text{old}},i,t}} g_{\theta,i,t}$ |
| AR-Lopti | $\frac{r_i - m}{|o_i|\sigma} \sum_{t=1}^{|o_i|} \left[ \eta\pi_{\theta_{\text{old}},i,t} + (1-\eta) \right] \frac{\pi_{\theta,i,t}}{\pi_{\theta_{\text{old}},i,t}} g_{\theta,i,t}$ |
| GPG | $\frac{1}{\sum_{j=1}^{G} |o_j|}(r_i - m) \left( \sum_{t=1}^{|o_i|} g_{\theta,i,t} \right)$ |
| GSPO | $\frac{c_{\theta,i}(r_i - m)}{c_{\theta_{\text{old}},i}\sigma} \left( \frac{1}{|o_i|} \sum_{t=1}^{|o_i|} g_{\theta,i,t} \right)$ |
| C$^2$GSPG | $\left( \frac{r_i - m}{1 - c_{\theta_{\text{old}},i}} + \frac{\beta(r_i - c_{\theta,i})}{1 - c_{\theta,i}} \right) \left( \frac{1}{|o_i|} \sum_{t=1}^{|o_i|} g_{\theta,i,t} \right)$ |

**1) The elimination of token-level bias.** Proposition 4.2 in (Yang et al., 2025b) shows that the gradient of GRPO suffers the over-dominance of low-probability tokens. AR-Lopti (Yang et al., 2025b) introduces token-specific adaptive weights to mitigate this issue. The sequence-based policy optimization methods, including GPG (Chu et al., 2025), GSPO (Zheng et al., 2025), and our C$^2$GSPG, can alleviate the token-level bias by computing a single policy gradient signal for an entire sequence. This setting treats different tokens evenly, which promotes a more stable and uniform training process.

**2) The mitigation of sequence-level bias.** Our method provides a more robust mechanism for leveraging model confidence. As shown in Table 2, the advantage term (i.e., $r_i - m$) of GSPO is modulated by $c_{\theta,i}/(c_{\theta_{\text{old}},i}\sigma)$. It means that when the current policy $\pi_\theta$ has a high uncertainty on a question $q$ (i.e., $\sigma$ is large) and generates a low-confidence sequence $o_i$, GSPO tends to weaken the significance of $q$ and $o_i$ in the gradient. As a result, GSPO (Zheng et al., 2025) relies more on those high-confidence sequences during training and thus inevitably suffers from the overconfidence issue. On the contrary, C$^2$GSPG modulates the advantage term by $1/(1 - c_{\theta_{\text{old}},i})$. This weight is in the range $[1, \infty)$, avoiding the gradient diminishing of low-confidence sequences. Concurrently, we omit the variance-based normalization because this technique has recently been shown to produce uncalibrated confidence estimates (Bereket & Leskovec, 2025).

## 3.2 MITIGATING GRADIENT CONFLICTS IN NON-BINARY REWARD SCENARIOS

In Proposition 3.1, we establish that for binary reward scenarios, the policy gradient term and the confidence-based regularizer always share the same sign. However, this synergy does not inherently extend to scenarios with non-binary rewards (e.g., the rewards in the K&K dataset (Xie et al., 2024)

---

[1] When formulating the gradients, we ignore the KL-divergence term and the clipping function. Such a simplification does not impact the correctness of our analysis. For AR-Lopti, the hyperparameter $\eta \in [0, 1]$. For GSPO, the $s_{\theta,i}$ in (4) is equal to $c_{\theta,i}/c_{\theta_{\text{old}},i}$.

are $\{-3, -1, -0.5, 3\}$). In particular, a gradient conflict can arise when a reward $r_i$ falls between the mean reward $m$ and the model confidence $c_{\theta,i}$. To mitigate this risk, we apply a nonlinear normalization method to rewards and an adaptive clipping strategy to the regularizer weight $\beta$, respectively, reformulating the objective function of C$^2$GSPG as

$$\mathcal{J}_{\text{C}^2\text{GSPG}}(\theta) = \mathbb{E}_{\{o_i\} \sim \pi_{\theta_{\text{old}}}} \frac{1}{G} \sum_{i=1}^{G} \Big[ \tilde{A}_i \cdot \log c_{\theta,i} + \mathbb{I}_i(\beta) \cdot [\hat{r}_i \log c_{\theta,i} + (1 - \hat{r}_i) \log(1 - c_{\theta,i})] \Big],$$

(10)

where $\hat{r}_i = \sigma(\alpha \cdot r_i) = \dfrac{1}{1 + \exp(-\alpha \cdot r_i)}$, $\mathbb{I}_i(\beta) = \begin{cases} \beta, & \text{if } \text{sign}(\hat{r}_i - \hat{m}) = \text{sign}(\hat{r}_i - c_{\theta,i}), \\ 0, & \text{otherwise.} \end{cases}$

Here, $\hat{r}_i$ is a sigmoid function that makes a nonlinear normalization on the original rewards. This maps any arbitrary reward to a standardized range of $[0, 1]$, aligning its scale with that of the confidence score $c_{\theta,i}$. The $\alpha$ controls the steepness of the normalization. Accordingly, the advantage is computed based on the normalized reward, i.e., $\tilde{A}_i = (\hat{r}_i - \hat{m})/(1 - c_{\theta_{\text{old}},i})$, where the mean reward is $\hat{m} = \sum_{i=1}^{G} \hat{r}_i$. As illustrated in Figure 2, the sigmoid function is a monotonically increasing function, which maintains the order of the rewards. Moreover, by pushing positive and negative rewards towards 1 and 0, respectively, it amplifies the gap between positive and negative feedback and thus reduces the risk of gradient conflict. In addition, for the sequence whose normalized reward $\hat{r}_i$ is still between $\hat{m}$ and $c_{\theta,i}$, the adaptive clipping function $\mathbb{I}_i(\beta) = 0$, ensuring that a conflicting confidence adjustment never counteracts a valid policy signal. In Appendix A.2, we show the scheme of C$^2$GSPG in detail.

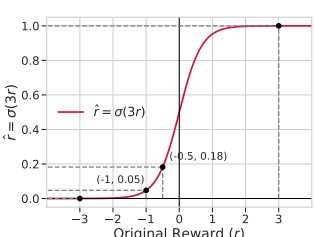

Figure 2: The normalization of the rewards in the K&K dataset (Xie et al., 2024).

## 4 EXPERIMENTS

To validate the effectiveness of C$^2$GSPG, we conduct experiments on logical and mathematical reasoning tasks. In each task, we compare C$^2$GSPG with state-of-the-art methods, including GRPO (Shao et al., 2024), GPG (Chu et al., 2025), GSPO (Zheng et al., 2025), and AR-Lopti (Yang et al., 2025b). **Representative experimental results are shown below, and more implementation details and results are given in the Appendix.**

### 4.1 IMPLEMENTATION DETAILS

For the logical reasoning task, we employ the K&K dataset (Xie et al., 2024), stratified by character count (3 to 7) to represent varying difficulty. Following the methodology of (Yang et al., 2025b), we select the Qwen2.5-3B-Instruct model[2] as our backbone and train it on a composite dataset encompassing all difficulty levels. Our training process is guided by a reward function adapted from Logic-RL (Xie et al., 2025), which combines structural and correctness signals. The raw discrete reward signal of $\{-3, -1, -0.5, 3\}$ is transformed through the proposed sigmoid mapping: boundary scores $-3$ and $3$ are mapped to $0$ and $1$, while intermediate values are converted via (10) (with $\alpha = 3$), yielding a final reward set of $\{0, 0.047426, 0.182426, 1\}$. The hyperparameter $\beta = 0.03$, and we pass it through the proposed clipping function $\mathbb{I}(\cdot)$.

For the mathematical reasoning task, we utilize the Qwen2.5-Math-1.5B model[3], which we train on the DeepScaleR-Uniform dataset (Shi et al., 2025), a curated subset of the DeepScaleR benchmark (Luo et al., 2025). Our training methodology employs a straightforward binary reward mechanism: a reward of $+1$ is assigned if the model's final answer is correct, and $0$ otherwise. In this case, we set $\beta \in \{0.01, 0.1, 0.5\}$. To assess generalization, we evaluate performance across a diverse suite of benchmarks, including GSM8K (Cobbe et al., 2021), Olympiad (He et al., 2024), Minerva (Lewkowycz et al., 2022), MATH 500 (Lightman et al., 2024), AMC2023, and AIME2024.

For each trained model $\pi_{\theta^*}$, the primary performance indicator is **accuracy**, calculated as the fraction of correctly answered questions. To further assess the reliability of the model's confidence

---

[2]The model is available at: `https://huggingface.co/Qwen/Qwen2.5-3B-Instruct`

[3]The model is available at: `https://huggingface.co/Qwen/Qwen2.5-Math-1.5B`

Table 3: The comparison for various methods on the K&K dataset. The row of "Base" shows the performance of Qwen2.5-3B-Instruct without post-training. In each column, the best result is **bold**.

| Method | Accuracy (%) by # Characters | | | | | Overall Performance | | |
|---|---|---|---|---|---|---|---|---|
| | 3 | 4 | 5 | 6 | 7 | Acc (↑) | BS (↓) | ECE (↓) |
| Base | 12.00 | 13.00 | 2.00 | 2.00 | 0.00 | 5.80 | 0.554 | 0.705 |
| GRPO | $58.67_{\pm4.20}$ | $44.93_{\pm3.99}$ | $34.53_{\pm2.69}$ | $31.87_{\pm2.92}$ | $20.53_{\pm3.01}$ | $38.11_{\pm2.68}$ | $0.405_{\pm0.018}$ | $0.497_{\pm0.023}$ |
| GPG | $69.53_{\pm17.58}$ | $64.80_{\pm25.28}$ | $49.47_{\pm28.39}$ | $43.20_{\pm24.87}$ | $35.93_{\pm21.37}$ | $52.59_{\pm22.67}$ | $0.363_{\pm0.230}$ | $0.363_{\pm0.242}$ |
| GSPO | $71.13_{\pm12.63}$ | $66.40_{\pm15.61}$ | $55.40_{\pm11.95}$ | $48.20_{\pm9.89}$ | $40.07_{\pm11.04}$ | $56.24_{\pm11.67}$ | $0.291_{\pm0.102}$ | $0.302_{\pm0.106}$ |
| AR-Lopti | $73.20_{\pm9.45}$ | $64.93_{\pm16.05}$ | $54.13_{\pm16.13}$ | $49.47_{\pm17.04}$ | $41.33_{\pm19.27}$ | $56.61_{\pm15.42}$ | $0.282_{\pm0.101}$ | $0.343_{\pm0.128}$ |
| C$^2$GSPG | $\textbf{86.00}_{\pm5.53}$ | $\textbf{88.00}_{\pm4.99}$ | $\textbf{78.60}_{\pm6.26}$ | $\textbf{73.27}_{\pm10.53}$ | $\textbf{69.00}_{\pm9.45}$ | $\textbf{78.97}_{\pm7.19}$ | $\textbf{0.136}_{\pm0.046}$ | $\textbf{0.126}_{\pm0.053}$ |

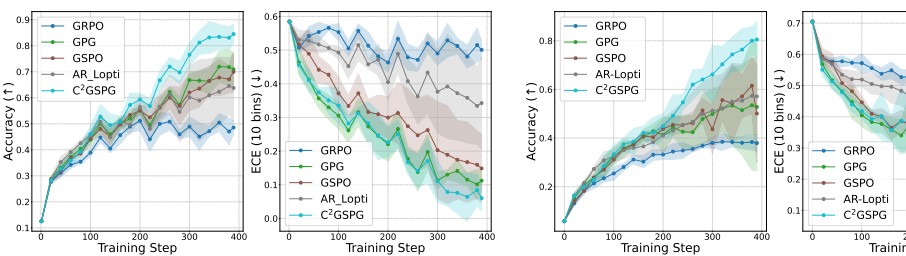

(a) The model performance on the training set      (b) The model performance on the testing set

Figure 3: Training dynamics on the K&K dataset. The right shows the evolution of Accuracy and ECE on the training set. The left shows the same metrics on the test set.

scores, we measure its calibration by computing the **Brier Score (BS)** (Glenn et al., 1950) and the **Expected Calibration Error (ECE)** (Naeini et al., 2015), which are defined as

$$\text{BS} = \frac{1}{N}\sum_{i=1}^{N}(c_{\theta^*,i} - r_i)^2, \quad \text{ECE} = \sum_{m=1}^{M}\frac{|B_m|}{N}\left|\text{acc}(B_m) - \text{conf}(B_m)\right|. \tag{11}$$

According to the definition, BS quantifies the mean squared difference between the predicted confidence $c_{\theta^*,i}$ and the actual outcome $r_i$ over all $N$ samples. ECE measures the discrepancy between confidence and accuracy by grouping predictions into $M$ bins. It is the weighted average of the absolute difference between the average accuracy and average confidence in each bin $B_m$. In our experiments, we set the number of bins to $M = 10$. For each evaluation metric, we run each method in five trials and record the mean and standard deviation of its performance.

## 4.2 LOGICAL REASONING

Table 3 summarizes the performance of various methods on the K&K dataset, with a detailed breakdown by difficulty level (number of characters) and overall calibration metrics. In terms of overall performance, C$^2$GSPG achieves an average accuracy that is over 22% higher than the strongest baseline, AR-Lopti, and the accuracy improvement is complemented by lower BS and ECE values. It means that C$^2$GSPG not only makes more accurate predictions but also produces more reliable confidence estimates. Furthermore, while all methods exhibit performance degradation as puzzle difficulty increases, C$^2$GSPG shows superior robustness. For instance, its performance lead over AR-Lopti widens from approximately 13% on 3-character puzzles to nearly 28% on the most challenging 7-character puzzles. This highlights the enhanced capability of our method to handle more complex reasoning tasks. Finally, the consistently small standard deviations across different runs underscore the stability and reliability of our approach.

In addition, we provide a quantitative and qualitative visualization analysis of different methods. In Figure 3, we illustrate the evolution of Accuracy and ECE on both the training and test sets. These plots show that C$^2$GSPG consistently achieves higher accuracy and lower ECE across both training and test sets, demonstrating its strong data fitting ability and generalization power.

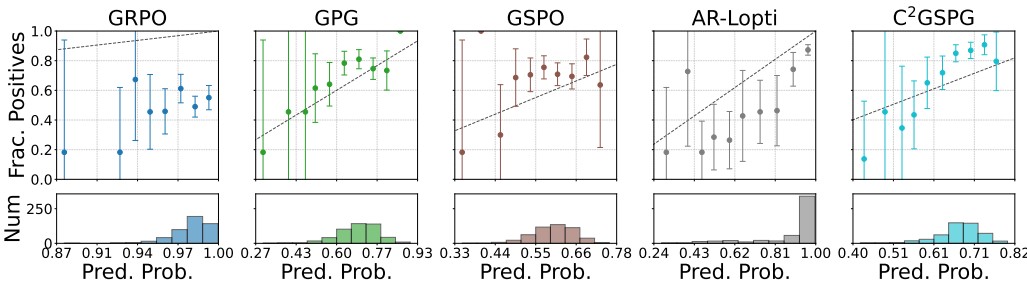

Figure 4: Calibration performance of various methods on the K&K dataset (test set). The top plot shows the Fraction of Positives against the model's predicted confidence, with the dashed line representing perfect calibration. The confidence histogram (bottom) shows the distribution of predicted confidences. Results are generated using a sampling strategy with a temperature of 1.0 and allowing the entire probability distribution to be considered.

Table 4: Ablation studies on the K&K dataset. We compare different variants of C$^2$GSPG with an adaptively-clipped or fixed regularization weight and a linear or sigmoid-based reward normalization. Here, the linear reward normalization is achieved by $\hat{r}_i = (r_i - r_{\min})/(r_{\max} - r_{\min})$.

| Method | | Accuracy (%) by # Characters | | | | | Overall Performance | | |
|---|---|---|---|---|---|---|---|---|---|
| $\mathbb{I}(\cdot)$ | $\sigma(\cdot)$ | 3 | 4 | 5 | 6 | 7 | Acc (↑) | BS (↓) | ECE (↓) |
| × | Linear | $77.93_{\pm 3.64}$ | $75.47_{\pm 7.95}$ | $63.13_{\pm 12.16}$ | $53.60_{\pm 12.13}$ | $44.93_{\pm 12.36}$ | $63.01_{\pm 9.36}$ | $0.229_{\pm 0.028}$ | $0.226_{\pm 0.040}$ |
| ✓ | Linear | $79.33_{\pm 5.67}$ | $79.87_{\pm 7.20}$ | $70.07_{\pm 6.69}$ | $64.00_{\pm 9.15}$ | $57.20_{\pm 10.33}$ | $70.09_{\pm 7.60}$ | $0.169_{\pm 0.059}$ | $0.171_{\pm 0.048}$ |
| × | Sigmoid | $78.93_{\pm 6.09}$ | $79.13_{\pm 11.17}$ | $66.33_{\pm 15.14}$ | $60.33_{\pm 18.08}$ | $56.53_{\pm 19.05}$ | $68.25_{\pm 13.86}$ | $0.216_{\pm 0.110}$ | $0.208_{\pm 0.112}$ |
| ✓ | Sigmoid | $\mathbf{86.00_{\pm 5.53}}$ | $\mathbf{88.00_{\pm 4.99}}$ | $\mathbf{78.60_{\pm 6.26}}$ | $\mathbf{73.27_{\pm 10.53}}$ | $\mathbf{69.00_{\pm 9.45}}$ | $\mathbf{78.97_{\pm 7.19}}$ | $\mathbf{0.136_{\pm 0.046}}$ | $\mathbf{0.126_{\pm 0.053}}$ |

To demonstrate that C$^2$GSPG helps calibrate model confidence, we compare it with baseline methods on their calibration curves and the histograms of confidence scores. The top plots in Figure 4 show that the calibration curve of C$^2$GSPG aligns closely to the perfect-calibration diagonal. In contrast, the baseline methods either result in a calibration curve falling below the diagonal (e.g., GRPO and AR-Lopti) or exhibit undesired high variance for their high-confidence sequences (e.g., GSPO), indicating a tendency towards overconfidence. In addition, some baselines fail to exhibit clear monotonically increasing curves, suggesting they do not effectively differentiate between levels of confidence when answering questions. The histograms of various methods (the bottom plots in Figure 4) further show that C$^2$GSPG produces a more evenly spread confidence distribution, while GRPO and AR-Lopti display a highly concentrated one with predictions clustered at the extremes.

**Ablation Studies.** To validate the rationality of our method, we conduct a series of ablation studies in Table 4, quantitatively analyze the impacts of adaptive regularizer clipping and sigmoid-based reward normalization on model performance. For the K&K dataset, given a set of sequences and their rewards, we apply C$^2$GSPG under different configurations. The results in Table 4 demonstrate that both the adaptive clipping mechanism and the sigmoid-based normalization help improve model performance consistently, leading to higher accuracy and better confidence calibration performance. These results demonstrate the necessity of mitigating gradient conflict in non-binary reward scenarios. In addition, the results in Tables 3 and 4 show that the simplified C$^2$GSPG, with fixed regularization weight and linear reward normalization, can still outperform all four baseline methods. This phenomenon indicates the effectiveness of our group sequence policy gradient framework.

### 4.3 MATHEMATICAL REASONING

Table 5 presents the results on the mathematical reasoning task. We compare C$^2$GSPG with varying values of $\beta$ against the baselines. Our results highlight a tunable trade-off between accuracy and calibration, governed by the regularization strength $\beta$. With a lower $\beta$, our approach surpasses baselines in terms of accuracy. As $\beta$ increases, the focus shifts towards calibration, resulting in superior BS and ECE at the cost of a modest dip in accuracy. The calibration results in Figure 1(a) further verify our above claim. The calibration curve for C$^2$GSPG is distinguished by its close alignment

Table 5: The comparison for various methods on mathematical reasoning benchmarks. The subscript in $C^2GSPG$ indicates the value of $\beta$. The row of "Base" shows the performance of Qwen2.5-Math-1.5B without post-training. In each column, the best result is **bold**.

| Method | Testing Data Accuracy (%) | | | | | | Overall Performance | | |
|---|---|---|---|---|---|---|---|---|---|
| | GSM8K | MATH500 | AMC2023 | AIME2024 | Olympiad | Minerva | Acc (↑) | BS (↓) | ECE (↓) |
| Base | 67.63 | 58.00 | 44.69 | 6.25 | 19.11 | 9.93 | 34.27 | 0.555 | 0.575 |
| GRPO | $79.57_{\pm0.62}$ | $69.92_{\pm0.48}$ | $54.67_{\pm2.67}$ | $13.06_{\pm3.02}$ | $23.47_{\pm0.45}$ | $15.88_{\pm0.30}$ | $42.76_{\pm0.96}$ | $0.491_{\pm0.008}$ | $0.501_{\pm0.010}$ |
| GPG | $79.85_{\pm0.51}$ | $69.15_{\pm0.93}$ | $56.81_{\pm2.99}$ | $13.94_{\pm2.50}$ | $22.77_{\pm0.32}$ | $15.64_{\pm1.45}$ | $43.03_{\pm0.98}$ | $0.489_{\pm0.010}$ | $0.498_{\pm0.011}$ |
| GSPO | $80.30_{\pm0.34}$ | $69.87_{\pm0.59}$ | $52.60_{\pm2.46}$ | $13.64_{\pm1.49}$ | $23.18_{\pm0.41}$ | $16.50_{\pm0.21}$ | $42.68_{\pm0.43}$ | $0.489_{\pm0.005}$ | $0.498_{\pm0.005}$ |
| AR-Lopti | $79.91_{\pm0.58}$ | $69.76_{\pm0.47}$ | $\mathbf{57.08_{\pm1.76}}$ | $15.08_{\pm2.39}$ | $\mathbf{24.07_{\pm0.32}}$ | $17.08_{\pm0.66}$ | $43.83_{\pm0.58}$ | $0.468_{\pm0.005}$ | $0.478_{\pm0.006}$ |
| $C^2GSPG_{0.01}$ | $\mathbf{82.72_{\pm0.15}}$ | $\mathbf{72.15_{\pm0.75}}$ | $54.25_{\pm2.61}$ | $\mathbf{16.33_{\pm2.13}}$ | $23.86_{\pm0.41}$ | $18.36_{\pm0.72}$ | $\mathbf{44.61_{\pm0.62}}$ | $0.484_{\pm0.005}$ | $0.491_{\pm0.006}$ |
| $C^2GSPG_{0.1}$ | $82.66_{\pm0.51}$ | $70.97_{\pm0.26}$ | $54.37_{\pm2.13}$ | $15.39_{\pm1.83}$ | $23.56_{\pm0.41}$ | $\mathbf{18.58_{\pm0.75}}$ | $44.25_{\pm0.51}$ | $0.473_{\pm0.004}$ | $0.481_{\pm0.005}$ |
| $C^2GSPG_{0.5}$ | $82.29_{\pm0.37}$ | $71.07_{\pm0.53}$ | $52.88_{\pm3.41}$ | $13.75_{\pm2.01}$ | $23.26_{\pm0.23}$ | $17.38_{\pm1.02}$ | $43.44_{\pm0.67}$ | $\mathbf{0.463_{\pm0.007}}$ | $\mathbf{0.464_{\pm0.007}}$ |

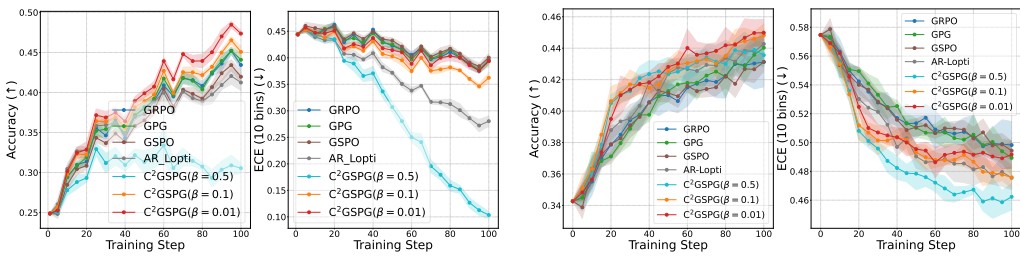

(a) The model performance on the training set      (b) The model performance on the testing set

Figure 5: Training dynamics on the mathematical dataset. The right shows the evolution of Accuracy and ECE on the training set. The left shows the same metrics on the test set.

with the ideal diagonal and its consistent positive slope. In contrast, all other baseline methods suffer from severe overconfidence, with their calibration curves falling far below the diagonal. Figure 5 visualizes the training dynamics of different methods and the impact of $\beta$ on our method.

It should be noted that the trade-off shown in the results is not in conflict with Proposition 3.1. In binary reward scenarios, the regularizer and the policy optimization term have the same gradient direction indeed. In other words, the regularizer implicitly increases the learning rate according to the consistency between the confidence score and the reward of each sequence. As a result, an improperly large $\beta$ may lead to an over-aggressive step size during training and thus result in performance degradation.

## 4.4 FURTHER ANALYSIS

The results in Tables 3 and 5 indicate that our method achieves significant improvements in logical reasoning but incremental gains in mathematical reasoning. In addition, when changing $\beta$, the reasoning accuracy and confidence calibration metrics (BS and ECE) show a trade-off on the math datasets, whereas improvements in these metrics are consistent across the K&K dataset, as shown in Table 10 in the Appendix. Note that these two phenomena are common (Zhang et al., 2025; Lacombe et al., 2025; Yuan et al., 2025), whose reasons are the difficulties of the reasoning tasks and the capabilities of the pre-trained LLMs w.r.t. the tasks.

- When the pre-trained LLM is strong while the task is relatively easy (e.g., K&K), the LLM possesses sufficient latent capacity for the task. Thus, fine-tuning with different methods can yield substantial performance gains. Notably, since K&K does not appear in the pre-training corpora of mainstream LLMs, these performance improvements primarily stem from genuine learning rather than data memorization. Consequently, the performance improvement of our method compared to other baselines further validates its effectiveness.

- When the pre-trained LLM is not strong enough while the task is challenging (e.g., the mathematical datasets), the work in (Yue et al., 2025) points out that although RL methods can improve sample efficiency, they remain unable to exceed the upper bound of their capabilities, thus resulting in limited performance gains. In such a situation, an overconfident model is encouraged by RL methods (including ours) because overconfident guessing may yield higher rewards, whereas an honest response often leads to failure. To maximize rewards, the model resorts to strategic guessing, leading to improved accuracy but degraded confidence calibration.

To address the above issue, we will further explore how to improve the calibration of models whose capabilities are currently insufficient for handling challenging tasks.

## 5 Conclusion

In this work, we propose $C^2$GSPG, a confidence-calibrated group sequence policy gradient method designed to suppress the overconfidence issue in reinforcement learning-based reasoning models. Within the group sequence policy gradient framework, $C^2$GSPG introduces a confidence-modulated advantage and confidence calibration regularizer, which effectively eliminates token-level bias and enforces consistency between model confidence and rewards. Experimental results on logical and mathematical reasoning tasks demonstrate that $C^2$GSPG significantly improves both reasoning accuracy and confidence calibration, outperforming existing state-of-the-art methods.

**Limitations and Future Work.** On the mathematical reasoning task, we observe that the gap in confidence calibration between the training and test sets remains relatively large, which may stem from the distribution discrepancy between the two datasets. Additionally, the model performance is significantly impacted by the choice of $\beta$. Addressing these issues, we plan to explore methods to enhance out-of-distribution robustness and calibration, as well as develop more effective regularization strategies to suppress overconfidence in erroneous generations.

## Ethics Statement

The datasets used in this paper are all publicly available and do not involve any ethical issues.

## Reproducibility Statement

We are committed to ensuring the reproducibility of our research. The source code is provided in the supplementary materials, which can be used to reproduce the main results presented in this paper.

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

APPENDIX

# A DETAILS OF OUR METHOD

## A.1 THE GRADIENT DERIVATION OF OUR METHOD

We will give a detailed derivation of our C$^2$GSPG method. First, we have

$$\nabla_\theta \, c_{\theta,i} = c_{\theta,i} \cdot \Big( \frac{1}{|o_i|} \sum\nolimits_{t=1}^{|o_i|} \nabla_\theta \log \pi_{\theta,i,t} \Big), \text{ and } \nabla_\theta \, \log c_{\theta,i} = \Big( \frac{1}{|o_i|} \sum\nolimits_{t=1}^{|o_i|} \nabla_\theta \log \pi_{\theta,i,t} \Big). \quad (12)$$

Accordingly, we have

$$\nabla_\theta \mathcal{J}_{\text{C}^2\text{GSPG}}(\theta)$$

$$= \nabla_\theta \mathbb{E}_{\{o_i\} \sim \pi_{\theta_{\text{old}}}} \left[ \frac{1}{G} \sum_{i=1}^{G} \Big( \tilde{A}_i \log(c_{\theta,i}) + \beta \left( r_i \log(c_{\theta,i}) + (1 - r_i) \log(1 - c_{\theta,i}) \right) \Big) \right]$$

$$= \mathbb{E}_{\{o_i\} \sim \pi_{\theta_{\text{old}}}} \left[ \frac{1}{G} \sum_{i=1}^{G} \Big( \tilde{A}_i \nabla_\theta \log(c_{\theta,i}) + \beta r_i \nabla_\theta \log(c_{\theta,i}) + \beta(1 - r_i) \nabla_\theta \log(1 - c_{\theta,i}) \Big) \right]$$

$$= \mathbb{E}_{\{o_i\} \sim \pi_{\theta_{\text{old}}}} \left[ \frac{1}{G} \sum_{i=1}^{G} \Big( (\tilde{A}_i + \beta r_i) \nabla_\theta \log(c_{\theta,i}) + \beta(1 - r_i) \Big( \frac{-\nabla_\theta c_{\theta,i}}{1 - c_{\theta,i}} \Big) \Big) \right]$$

$$= \mathbb{E}_{\{o_i\} \sim \pi_{\theta_{\text{old}}}} \left[ \frac{1}{G} \sum_{i=1}^{G} \Big( (\tilde{A}_i + \beta r_i) \nabla_\theta \log(c_{\theta,i}) - \beta(1 - r_i) \frac{c_{\theta,i} \cdot \nabla_\theta \log(c_{\theta,i})}{1 - c_{\theta,i}} \Big) \right] \quad (13)$$

$$= \mathbb{E}_{\{o_i\} \sim \pi_{\theta_{\text{old}}}} \left[ \frac{1}{G} \sum_{i=1}^{G} \Big( \tilde{A}_i + \beta r_i - \beta(1 - r_i) \frac{c_{\theta,i}}{1 - c_{\theta,i}} \Big) \nabla_\theta \log(c_{\theta,i}) \right]$$

$$= \mathbb{E}_{\{o_i\} \sim \pi_{\theta_{\text{old}}}} \left[ \frac{1}{G} \sum_{i=1}^{G} \Big( \tilde{A}_i + \beta \frac{r_i(1 - c_{\theta,i}) - (1 - r_i)c_{\theta,i}}{1 - c_{\theta,i}} \Big) \nabla_\theta \log(c_{\theta,i}) \right]$$

$$= \mathbb{E}_{\{o_i\} \sim \pi_{\theta_{\text{old}}}} \left[ \frac{1}{G} \sum_{i=1}^{G} \Big( \tilde{A}_i + \beta \frac{r_i - c_{\theta,i}}{1 - c_{\theta,i}} \Big) \nabla_\theta \log(c_{\theta,i}) \right]$$

$$= \mathbb{E}_{\{o_i\} \sim \pi_{\theta_{\text{old}}}} \left[ \frac{1}{G} \sum_{i=1}^{G} \Big( \frac{r_i - m}{1 - c_{\theta_{\text{old}},i}} + \beta \frac{r_i - c_{\theta,i}}{1 - c_{\theta,i}} \Big) \Big( \frac{1}{|o_i|} \sum_{t=1}^{|o_i|} \nabla_\theta \log \pi_{\theta,i,t} \Big) \right].$$

In our method, the reason that we use the binary cross-entropy (BCE) based regularizer rather than mean squared error (MSE) for calibrating model confidence is shown below. The objective function with MSE loss is:

$$\mathcal{J}_{\text{C}^2\text{GSPG}}^{\text{MSE}}(\theta) = \mathbb{E}_{\{o_i\} \sim \pi_{\theta_{\text{old}}}} \frac{1}{G} \sum_{i=1}^{G} \Big[ \tilde{A}_i \cdot \log c_{\theta,i} - \beta(r_i - c_{\theta,i})^2 \Big], \quad (14)$$

The gradient of it is:

$$\nabla_\theta \mathcal{J}_{\text{C}^2\text{GSPG}}^{\text{MSE}}(\theta)$$

$$= \mathbb{E}_{\{o_i\} \sim \pi_{\theta_{\text{old}}}} \frac{1}{G} \sum_{i=1}^{G} \Big( \tilde{A}_i - 2\beta \cdot c_{\theta,i}(c_{\theta,i} - r_i) \Big) \Big( \frac{1}{|o_i|} \sum_{t=1}^{|o_i|} \nabla_\theta \log \pi_{\theta,i,t} \Big) \quad (15)$$

$$= \mathbb{E}_{\{o_i\} \sim \pi_{\theta_{\text{old}}}} \frac{1}{G} \sum_{i=1}^{G} \Big( \frac{r_i - m}{1 - c_{\theta_{\text{old}},i}} - 2\beta \cdot c_{\theta,i}(c_{\theta,i} - r_i) \Big) \Big( \frac{1}{|o_i|} \sum_{t=1}^{|o_i|} \nabla_\theta \log \pi_{\theta,i,t} \Big).$$

From (15), we see that the MSE regularizer scales the gradient directly with $c_{\theta,i}(r_i - c_{\theta,i})$. For sequences with low predicted confidence, this results in a small gradient, producing a weak corrective signal. In contrast, as mentioned in Section 3.1, BCE introduces a $1/(1 - c_{\theta,i})$ modulation, which avoid this problem.

## A.2 EXPERIMENTAL DETAILS

All experiments are performed using the VERL framework[4] on NVIDIA A100 GPUs. The whole training algorithm of our method is shown in Algorithm 1.

---

**Algorithm 1** The C$^2$GSPG Training Algorithm

---

**Require:** Initial policy $\pi_\theta = \pi_{\text{ref}}$, dataset $\mathcal{D}$.

1: **for** each training epoch **do**
2:     **for** each RL step, Sample prompts $\{q\}^M \sim \mathcal{D}$ **do**
3:         Sample $G$ responses $\{o_i\}_{i=1}^G \sim \pi_{\theta_{\text{old}}}(\cdot|q)$.
4:         For each response, calculate reward $r_i$ and confidence $c_{\theta_{\text{old}},i}$.
5:         **if** binary reward **then**
6:             Compute advantage $\tilde{A}_i = \frac{r_i - m}{1 - c_{\theta_{\text{old}},i}}$, where $m = \frac{1}{G}\sum_{i=1}^G r_i$.
7:         **else**
8:             Normalize reward: $\hat{r}_i = \sigma(\alpha \cdot r_i) = \frac{1}{1 + \exp(-\alpha \cdot r_i)}$.
9:             Compute average normalized reward $\hat{m} = \frac{1}{G}\sum_{i=1}^G \hat{r}_i$.
10:           Compute advantage $\tilde{A}_i = \frac{\hat{r}_i - \hat{m}}{1 - c_{\theta_{\text{old}},i}}$.
11:         **end if**
12:         **for** each RL epoch, Sample a mini-batch from rollouts **do**
13:             Compute confidence $c_{\theta,i}$ with current policy $\pi_\theta$.
14:             **if** non binary reward **then**
15:                 Determine $\mathbb{I}_i(\beta) = \begin{cases} \beta, & \text{if } \text{sign}(\hat{r}_i - \hat{m}) = \text{sign}(\hat{r}_i - c_{\theta,i}), \\ 0, & \text{otherwise.} \end{cases}$
16:             **end if**
17:             Update policy parameters $\theta$ using (6) (binary) or (10) (non binary).
18:         **end for**
19:     **end for**
20: **end for**
21: **return** Final policy parameters $\theta^*$.

---

The key hyperparameters for our method are detailed in Table 6. To generate settings, we exclude the clipping mechanism in GSPO and fix the $F_{norm}$ parameter to 1 in GPG. For the mathematical reasoning task, the model is trained for 15 epochs (approximately 100 steps). The final reported performance is the average of scores from the 80th, 90th, and 100th steps. Furthermore, to mitigate variance on the smaller AMC 2023 and AIME 2024 test sets, we report the average performance over 8 independent runs for these specific benchmarks. For the logical reasoning task, the model is trained for 5 epochs for all methods. To assess performance, we evaluate the model every 20 steps on separate test sets for each level, and the final reported score is the average of the last three evaluation checkpoints. Furthermore, the low clip ratio and high clip ratio in GRPO and AR-Lopti are specifically set to 0.20 and 0.24 following (Yang et al., 2025b) on this task. Moreover, we apply a KL divergence penalty (with weight $\gamma = 0.001$) to all baselines on the logical reasoning task. For the mathematical reasoning task, we omit it for our method and GPG, but retain it for all other baselines with the same coefficient. The other hyperparameters are mostly set as the default values in VERL.

**Greedy search v.s. sampling for sequence generation and evaluation.** For the results reported in Tables 3- 5 we use greedy decoding to generate sequences. Accuracy, Expected Calibration Error (ECE), and Brier Score (BS) are computed based on these greedy outputs and their corresponding model confidence scores. For Figures 4 and 1(a), we use a sampling-based decoding strategy with temperature equals 1.0. This allows visualization of model calibration across a broader confidence range, including low-confidence predictions that are less frequent under greedy decoding. So, we can have a more comprehensive view of model calibration behavior.

---

[4]The VERL framework is available at: `https://github.com/volcengine/verl`

Table 6: Comparison of Key Hyperparameters for Mathematical and Logical Reasoning Tasks

| Parameter | Mathematical | Logic |
|---|---|---|
| Base Model | Qwen2.5-Math-1.5B | Qwen2.5-3B-Instruct |
| max_prompt_length | 1024 | 400 |
| max_response_length | 3000 | 4096 |
| total_epochs | 15 | 5 |
| learning rate | 1e-6 | 1e-6 |
| train_batch_size | 1024 | 64 |
| ppo_mini_batch_size | 256 | 256 |
| ppo_micro_batch_size | 16 | 32 |
| kl_loss_coef | 0 | 0.001 |
| rollout_n | 4 | 8 |
| rollout_temperature | 1.0 | 0.7 |
| beta | 0.5 | 0.03 |
| clip method | no clip | beta clip |
| n_gpus | 8 | 4 |
| test_freq | 5 | 20 |

## B  More Experiment Results

### B.1  Results on Mathematical Reasoning task

Figure 6(a) reports the calibration performance on the mathematical training set, which can conclude similar results with Figure 1(a). All methods produce monotonically increasing Fraction of Positives curves (top) with respect to predicted probabilities. However, the curve of our method lies closest to the perfect-calibration diagonal, whereas those of the baselines consistently fall below it, indicating that they tend to be overconfident. Furthermore, the accompanying histogram of predicted probabilities (bottom) shows that our method yields a more dispersed distribution. This suggests a superior ability to differentiate confidence levels, assigning nuanced probabilities rather than collapsing towards extreme values.

### B.2  Results on Logical Reasoning task

Figure 6(b) presents the calibration dynamics of the models on the K&K training set. In conjunction with Figure 4, we observe that the calibration curves on both the training and test sets are highly consistent, exhibiting a monotonic increasing trend and remaining close to the perfect-calibration diagonal. These results indicate that our method effectively mitigates the issue of model overconfidence while achieving strong generalization performance.

### B.3  Reward Formulation for Logical Reasoning

The K&K logic puzzles dataset is based on the classic "Knights and Knaves" scenario. In this setting, each inhabitant of an island is either a Knight (who always tells the truth) or a Knave (who always lies). The model's objective is to deduce the true identity of each character from their statements.

The complexity of this task is reflected in its unique reward structure, which generates a composite score from the set $\{-3, -1, -0.5, 3\}$. This score is calculated based on two distinct criteria:

- **Formatting Correctness:** The model is required to produce a Chain-of-Thought (CoT) rationale enclosed within `<think>` tags, followed by the final solution in `<answer>` tags. Adherence to this format yields +1 point, while any deviation results in a penalty of -1 point.

- **Answer Accuracy:** The final answer is graded on its correctness. A completely correct solution receives +2 points. A partially correct answer incurs a penalty of -1.5 points, and a fully incorrect answer is penalized by -2 points.

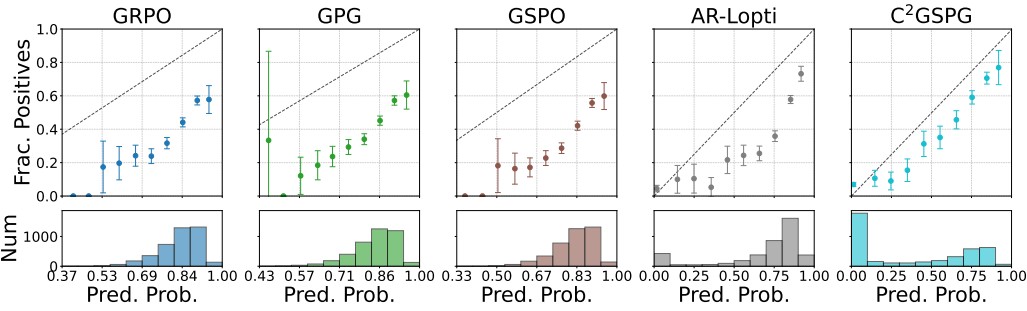

(a) Calibration performance of various methods on mathematical reasoning training set

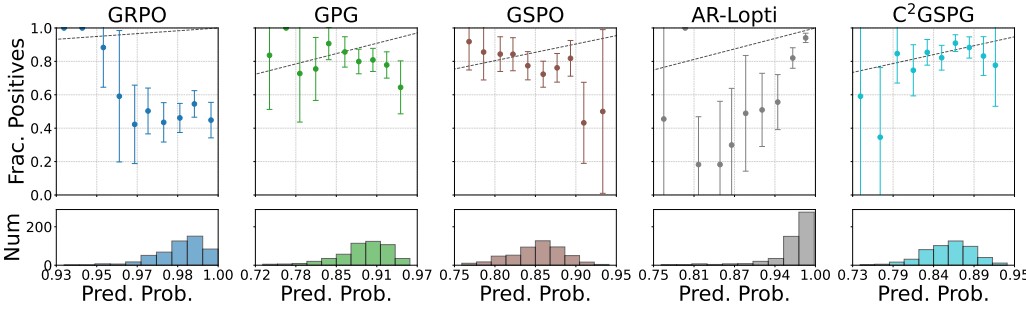

(b) Calibration performance of various methods on K&K training set

Figure 6: Calibration performance of various methods on the mathematical reasoning and K&K training dataset. Each panel consists of two subplots. The top plot shows the Fraction of Positives against the model's predicted confidence, with the dashed line representing perfect calibration. The confidence histogram (bottom) illustrates the distribution of predicted confidence levels.

The final reward is the sum of the scores from these two criteria. This multi-faceted reward signal is designed to evaluate both the model's reasoning process and the accuracy of its conclusions.

### B.4  CASE STUDIES IN MATHEMATICAL REASONING

---

## Case Study 1

**Question:** Alisa biked 12 miles per hour for 4.5 hours. Stanley biked at 10 miles per hour for 2.5 hours. How many miles did Alisa and Stanley bike in total? Let's think step by step and output the final answer within $\boxed{\text{answer}}$. (Ground-truth answer is 79)

**GRPO Answer** (Confidence: 0.9428): To find the total distance biked by Alisa and Stanley, we can follow these steps:

1. Calculate the distance biked by Alisa.

2. Calculate the distance biked by Stanley.

3. Add the two distances together to get the total distance.

The formula to calculate distance is:

$$\text{Distance} = \text{Speed} \times \text{Time}$$

Let's break it down:

1. Alisa's distance:

$$\text{Distance}_{\text{Alisa}} = 12\,\text{miles/hour} \times 4.5\,\text{hours}$$

---

2. Stanley's distance:

$$\text{Distance}_{\text{Stanley}} = 10 \text{ miles/hour} \times 2.5 \text{ hours}$$

3. Total distance:

$$\text{Total Distance} = \text{Distance}_{\text{Alisa}} + \text{Distance}_{\text{Stanley}}$$

The total distance biked by Alisa and Stanley is $\boxed{83}$ miles. The answer is: **83**

**GPG Answer** (Confidence: 0.9341): To find the total distance biked by Alisa and Stanley, we can follow these steps:

1. Calculate the distance biked by Alisa.
2. Calculate the distance biked by Stanley.
3. Add the two distances together to get the total distance.

The formula to calculate distance is:

$$\text{Distance} = \text{Speed} \times \text{Time}$$

Let's break it down:

1. Alisa's distance:

$$\text{Distance}_{\text{Alisa}} = 12 \text{ miles/hour} \times 4.5 \text{ hours}$$

2. Stanley's distance:

$$\text{Distance}_{\text{Stanley}} = 10 \text{ miles/hour} \times 2.5 \text{ hours}$$

3. Total distance:

$$\text{Total Distance} = \text{Distance}_{\text{Alisa}} + \text{Distance}_{\text{Stanley}}$$

The total distance biked by Alisa and Stanley is $\boxed{89}$ miles. The answer is: **89**

**GSPO Answer** (Confidence: 0.9162): To find the total distance biked by Alisa and Stanley, we can follow these steps:

1. Calculate the distance biked by Alisa.
2. Calculate the distance biked by Stanley.
3. Add the two distances together to get the total distance.

The formula to calculate distance is:

$$\text{Distance} = \text{Speed} \times \text{Time}$$

Let's break it down:

1. Alisa's distance:

$$\text{Distance}_{\text{Alisa}} = 12 \text{ miles/hour} \times 4.5 \text{ hours}$$

2. Stanley's distance:

$$\text{Distance}_{\text{Stanley}} = 10 \text{ miles/hour} \times 2.5 \text{ hours}$$

3. Total distance:

$$\text{Total Distance} = \text{Distance}_{\text{Alisa}} + \text{Distance}_{\text{Stanley}}$$

The total distance biked by Alisa and Stanley is $\boxed{83}$ miles. The answer is: **83**

**AR-Lopti Answer** (Confidence: 0.9074 ): To find the total distance biked by Alisa and Stanley, we can follow these steps:

    1. Calculate the distance biked by Alisa.

    2. Calculate the distance biked by Stanley.

    3. Add the two distances together to get the total distance.

The formula to calculate distance is:

$$\text{Distance} = \text{Speed} \times \text{Time}$$

Let's break it down:

    1. Alisa's distance:

$$\text{Distance}_{\text{Alisa}} = 12 \,\text{miles/hour} \times 4.5 \,\text{hours}$$

    2. Stanley's distance:

$$\text{Distance}_{\text{Stanley}} = 10 \,\text{miles/hour} \times 2.5 \,\text{hours}$$

    3. Total distance:

$$\text{Total Distance} = \text{Distance}_{\text{Alisa}} + \text{Distance}_{\text{Stanley}}$$

Now let's calculate these values using Python. The total distance biked by Alisa and Stanley is $\boxed{83}$ miles. The answer is: **83**

**C²GSPG Answer** (Confidence: 0.9086): First, we calculate the distance Alisa biked:

$$\text{Distance}_{\text{Alisa}} = \text{Speed}_{\text{Alisa}} \times \text{Time}_{\text{Alisa}} = 12 \,\text{miles/hour} \times 4.5 \,\text{hours}$$

Next, we calculate the distance Stanley biked:

$$\text{Distance}_{\text{Stanley}} = \text{Speed}_{\text{Stanley}} \times \text{Time}_{\text{Stanley}} = 10 \,\text{miles/hour} \times 2.5 \,\text{hours}$$

Then we add these distances together to find the total distance biked:

$$\text{Total Distance} = \text{Distance}_{\text{Alisa}} + \text{Distance}_{\text{Stanley}}$$

Now we perform the calculations:

$$\text{Distance}_{\text{Alisa}} = 12 \times 4.5 = 54 \,\text{miles}$$

$$\text{Distance}_{\text{Stanley}} = 10 \times 2.5 = 25 \,\text{miles}$$

$$\text{Total Distance} = 54 + 25 = 79 \,\text{miles}$$

So the total distance biked by Alisa and Stanley is $\boxed{79}$ miles. The answer is: **79**

## B.5 Case Studies in Logical Reasoning

### Case Study 1

**Question:** A very special island is inhabited only by knights and knaves. Knights always tell the truth, and knaves always lie. You meet 3 inhabitants: Emily, Aria, and Abigail. As Emily put it, "Emily is a knight or Aria is a knave". Aria said that "If Aria is a knight then Abigail is a knight". Abigail stated, "Aria is a knight if and only if Emily is a knave". So who is a knight and who is a knave? (Ground-truth answer is Emily is a knave. Aria is a knight. Abigail is a knight.)

**GRPO Answer** (Confidence: 0.9947): `<think>` Emily's statement is "Emily is a knight or Aria is a knave". If Emily were a knave, her statement would be false, but since she is a knight, her statement is true. This means that Emily is a knight. Since Emily is a knight, her statement "Emily is a knight or Aria is a knave" is true, so Aria must be a knight. Since Aria

is a knight, her statement "If Aria is a knight then Abigail is a knight" is true, so Abigail must be a knight. Finally, Abigail's statement is "Aria is a knight if and only if Emily is a knave". Since Aria is a knight and Emily is a knight, this statement is false, so Abigail is a knave. `</think>`
`<answer>` **Emily is a knight. Aria is a knight. Abigail is a knave.** `</answer>`

**GPG Answer** (Confidence: 0.9309): `<think>`Let's analyze the statements one by one and see if we can determine the nature of each inhabitant.

1. **Abigail's statement**: "Aria is a knight if and only if Emily is a knave."
   - If Abigail is a knight, then Aria must be a knave and Emily must be a knight. But this contradicts the statement that Aria is a knight if and only if Emily is a knave. So, Abigail must be a knave.
   - If Abigail is a knave, then Aria must not be a knight if Emily is a knave, and Aria must be a knight if Emily is a knight.
2. **Aria's statement**: "If Aria is a knight then Abigail is a knight."
   - Since we know Abigail is a knave, and Aria's statement is that if Aria is a knight, then Abigail is a knight, Aria must be a knave.
3. **Emily's statement**: "Emily is a knight or Aria is a knave."
   - Since Aria is a knave, the statement "Emily is a knight or Aria is a knave" is true. Therefore, Emily must be a knight.
4. **Aria's statement**: "If Aria is a knight then Abigail is a knight."
   - Since Aria is a knave, the statement "If Aria is a knight then Abigail is a knight" is vacuously true. So, Aria is a knave.
5. **Abigail's statement**: "Aria is a knight if and only if Emily is a knave."
   - Since Aria is a knave and Emily is a knight, Abigail's statement is true. Therefore, Abigail is a knight.
6. **Aria's statement**: "If Aria is a knight then Abigail is a knight."
   - Since Aria is a knave, the statement "If Aria is a knight then Abigail is a knight" is vacuously true. So, Aria is a knave.
7. **Emily's statement**: "Emily is a knight or Aria is a knave."
   - Since Emily is a knight and Aria is a knave, the statement is true. Therefore, Emily is a knight.

`</think>`
`<answer>` **Emily is a knight, Aria is a knave, Abigail is a knight.** `</answer>`

**GSPO Answer** (Confidence: 0.8906): `<think>`First, let's analyze Abigail's statement: "Aria is a knight if and only if Emily is a knave." This is a biconditional statement, so it must be either both true or both false. If Abigail were a knight, then this statement would be true, meaning Aria must be a knight and Emily must be a knave. However, if Emily were a knave, then the statement "Emily is a knight or Aria is a knave" would be true, which would mean Emily is a knight, creating a contradiction. Therefore, Abigail must be a knave.
Now, let's move to Aria's statement: "If Aria is a knight, then Abigail is a knight." Since Abigail is a knave, this conditional statement must be a true statement, meaning Aria must be a knight.
Next, let's examine Emily's statement: "Emily is a knight or Aria is a knave." Since we already know Aria is a knight, this statement must be true, so Emily must be a knight.
Finally, let's verify the consistency of these conclusions:

1. Emily is a knight (as verified by her statement).
2. Aria is a knight (as verified by her statement).
3. Abigail is a knave (as verified by her statement).

All statements are consistent with each other and with the rules of the island.
</think>
<answer> **Emily is a knight. Aria is a knight. Abigail is a knave.** </answer>

**AR-Lopti Answer** (Confidence: 0.9933): <think>Let's analyze the statements step by step:

1. Emily says, "Emily is a knight or Aria is a knave."

2. Aria says, "If Aria is a knight then Abigail is a knight."

3. Abigail says, "Aria is a knight if and only if Emily is a knave."

First, let's consider Emily's statement: "Emily is a knight or Aria is a knave." This is a tautology and must be true. Therefore, Emily must be a knight.
Now, let's look at Aria's statement: "If Aria is a knight then Abigail is a knight." This is a valid statement because if Aria is a knight, then Abigail must be a knight. Therefore, Aria must be a knight.
Finally, let's consider Abigail's statement: "Aria is a knight if and only if Emily is a knave." Since Emily is a knight, the condition "if and only if Emily is a knave" is false. Therefore, Abigail must be a knave.
So, we can conclude:

- Emily is a knight.

- Aria is a knight.

- Abigail is a knave.

Therefore, Emily is a knight, Aria is a knight, and Abigail is a knave.
</think>
<answer> **Emily is a knight. Aria is a knight. Abigail is a knave.** </answer>

**C²GSPG Answer** (Confidence: 0.9065): <think>Let's analyze the statements step by step. Let's analyze the statements step by step: 1. Emily says, "Emily is a knight or Aria is a knave." 2. Aria says, "If Aria is a knight then Abigail is a knight." 3. Abigail says, "Aria is a knight if and only if Emily is a knave."
First, let's consider Aria's statement. Aria's statement is a conditional statement: "If Aria is a knight, then Abigail is a knight." This is a true statement because any conditional statement "If P, then Q" is true unless P is true and Q is false. Since Aria is making a conditional statement, and Aria is a knight, the antecedent (Aria is a knight) is true, and the consequent (Abigail is a knight) must also be true for the whole statement to be true. Therefore, Abigail must be a knight.
Now let's consider Abigail's statement: "Aria is a knight if and only if Emily is a knave." Since we already determined that Abigail is a knight, the statement "Aria is a knight if and only if Emily is a knave" must be true. This means that if Aria is a knight, then Emily must be a knave, and if Emily is a knave, then Aria must be a knight.
Now let's consider Emily's statement: "Emily is a knight or Aria is a knave." Since Emily is a knave and Aria is a knight, the disjunction "Emily is a knight or Aria is a knave" is false. Therefore, Emily's statement is false, which means Emily is a knave.
In conclusion:

- Emily is a knave.

- Aria is a knight.

- Abigail is a knight.

</think>
<answer> **Emily is a knave. Aria is a knight. Abigail is a knight.** </answer>

## C   USE OF LARGE LANGUAGE MODELS (LLMS)

In this work, LLMs are used for language refinement of the paper and for assisting in verifying the correctness of code implementations.

# D  ADDED EXPERIMENT RESULTS

## D.1  SCALABILITY AND ROBUSTNESS ON LARGER MODELS

To further evaluate the scalability and robustness of our method, we conduct experiments on larger-scale models. Following Shi et al. (2025) and Yang et al. (2025b), we fine-tune the Qwen2.5-7B model[5] for the mathematical reasoning task, and the Qwen2.5-7B-Instruct-1M model[6] for the logical reasoning task, respectively. The batch size is halved due to computational constraints. For the mathematical reasoning task, training is slightly extended to 150 steps to better demonstrate robustness. Other settings remained unchanged. Results in Table 7 and Table 8 show that our method outperforms baselines consistently on larger LLMs, demonstrating its scalability.

Table 7: Comparison of various methods applied to 7B models on mathematical reasoning benchmarks. The row of "Base" shows the performance of the model without any post-training. In each column, the best result is highlighted in **bold**.

| Method | Accuracy (%) by Dataset | | | | | | Overall Performance | | |
|---|---|---|---|---|---|---|---|---|---|
| | GSM8K | MATH500 | AMC2023 | AIME2024 | Olympiad | Minerva | Acc ($\uparrow$) | BS ($\downarrow$) | ECE ($\downarrow$) |
| Base | 61.26 | 46.20 | 21.25 | 6.25 | 16.30 | 9.93 | 26.86 | 0.598 | 0.632 |
| GRPO | 89.60 | 72.04 | 43.87 | 16.33 | **25.10** | 23.82 | 45.13 | 0.531 | 0.535 |
| GPG | 89.63 | **73.24** | 51.56 | 13.00 | 24.62 | 23.75 | 45.97 | 0.515 | 0.520 |
| GSPO | 88.75 | 72.44 | **53.19** | 14.67 | 24.56 | 22.35 | 45.99 | 0.520 | 0.524 |
| AR-Lopti | 88.72 | 73.08 | 47.31 | 14.42 | 24.65 | **24.26** | 45.41 | 0.522 | 0.528 |
| $C^2$GSPG | **90.60** | 71.56 | 52.12 | **20.67** | 24.44 | 22.94 | **47.06** | **0.511** | **0.515** |

Table 8: Performance comparison of various methods on the K&K dataset for 7B models. The row of "Base" shows the performance of the model without post-training. In each column, the best result is highlighted in **bold**.

| Method | Accuracy (%) by # Characters | | | | | Overall Performance | | |
|---|---|---|---|---|---|---|---|---|
| | 3 | 4 | 5 | 6 | 7 | Acc ($\uparrow$) | BS ($\downarrow$) | ECE ($\downarrow$) |
| Base | 23.00 | 17.00 | 7.00 | 10.00 | 3.00 | 12.00 | 0.460 | 0.610 |
| GRPO | 79.00 | 72.40 | 56.40 | 56.60 | 42.40 | 61.36 | 0.254 | 0.308 |
| GPG | 90.60 | 88.20 | 73.20 | 69.60 | 53.60 | 75.04 | 0.190 | 0.194 |
| GSPO | 84.00 | 88.00 | 75.20 | 72.00 | 67.20 | 77.28 | 0.157 | 0.171 |
| AR-Lopti | 82.60 | 87.60 | 71.60 | 68.00 | 58.40 | 73.64 | 0.180 | 0.214 |
| $C^2$GSPG | **95.60** | **92.80** | **85.80** | **77.00** | **73.00** | **84.84** | **0.126** | **0.124** |

## D.2  ABLATION STUDY

**The Effect of Different Components in Our Method.** Our $C^2$GSPG method includes a GSPG framework and a BCE regularizer. To individually validate the effectiveness of these two modules, we design an ablation study on the K&K dataset using the Qwen2.5-3B-Instruct model. In particular, for each RL method (including baselines and our GSPG framework), we augment it by incorporating our BCE regularizer. The results in Table 9 show that a) the BCE regularizer is universally effective, which improves various methods consistently, and b) the GSPG framework outperforms the baselines consistently.

**The Effect of Hyperparameter $\beta$.** We fine-tune Qwen2.5-3B-Instruct on the K&K dataset with $\beta \in \{0.01, 0.03, 0.05\}$ to study its impact. The results in Table 10 show that an improperly set $\beta$ can lead to under- or over-regularization, and our setting ($\beta = 0.03$) leads to better performance. Notably, the models using sub-optimal settings ($\beta = 0.01$ and $0.05$) still outperform the baselines, demonstrating the robustness of our method.

---

[5]The model is available at: `https://huggingface.co/Qwen/Qwen2.5-7B`
[6]The model is available at: `https://huggingface.co/Qwen/Qwen2.5-7B-Instruct-1M`

Table 9: The comparison for various methods on the K&K dataset. The row of "Base" shows the performance of Qwen2.5-3B-Instruct without post-training. In each column, the best result is **bold**.

| Method | Accuracy (%) by # Characters | | | | | Overall Performance | | |
|---|---|---|---|---|---|---|---|---|
| | 3 | 4 | 5 | 6 | 7 | Acc ($\uparrow$) | BS ($\downarrow$) | ECE ($\downarrow$) |
| Base | 12.00 | 13.00 | 2.00 | 2.00 | 0.00 | 5.80 | 0.554 | 0.705 |
| GRPO | 58.67 | 44.93 | 34.53 | 31.87 | 20.53 | 38.11 | 0.405 | 0.497 |
| GRPO+BCE | 66.60 | 57.07 | 45.80 | 41.13 | 29.73 | 48.07 | 0.335 | 0.409 |
| GPG | 69.53 | 64.80 | 49.47 | 43.20 | 35.93 | 52.59 | 0.363 | 0.363 |
| GPG+BCE | 70.93 | 66.07 | 53.07 | 43.87 | 36.33 | 54.05 | 0.337 | 0.330 |
| GSPO | 71.13 | 66.40 | 55.40 | 48.20 | 40.07 | 56.24 | 0.291 | 0.302 |
| GSPO+BCE | 77.07 | 76.53 | 61.80 | 55.73 | 47.27 | 63.68 | 0.231 | 0.233 |
| AR-Lopti | 73.20 | 64.93 | 54.13 | 49.47 | 41.33 | 56.61 | 0.282 | 0.343 |
| AR-Lopti+BCE | 78.40 | 81.80 | 70.27 | 66.13 | 58.93 | 71.11 | 0.192 | 0.225 |
| GSPG | 79.50 | 75.50 | 67.00 | 57.00 | 50.00 | 65.80 | 0.215 | 0.256 |
| GSPG+BCE(Our C$^2$GSPG) | **86.00** | **88.00** | **78.60** | **73.27** | **69.00** | **78.97** | **0.136** | **0.126** |

Table 10: Ablation study on different values of $\beta$ on the K&K logic puzzles dataset.

| Beta ($\beta$) | Accuracy (%) by # Characters | | | | | Overall Performance | | |
|---|---|---|---|---|---|---|---|---|
| | 3 | 4 | 5 | 6 | 7 | Acc ($\uparrow$) | BS ($\downarrow$) | ECE ($\downarrow$) |
| 0.01 | 82.00 | 83.00 | 70.00 | 64.33 | 61.00 | 72.07 | 0.169 | 0.171 |
| 0.03 | 86.00 | 88.00 | 78.60 | 73.27 | 69.00 | 78.97 | 0.136 | 0.126 |
| 0.05 | 81.00 | 83.00 | 73.00 | 65.33 | 58.33 | 72.13 | 0.168 | 0.132 |

**The Effect of Hyperparameter $\alpha$.** We study the impact of the reward transformation hyperparameter, $\alpha \in \{1, 3, 5\}$, with results in Table 11. When $\alpha$ is too large, the reward transformation converges to a step function, reducing the distinguishability across different negative rewards. When $\alpha$ is too small, the negative reward after transformation is far from zero, leading to a higher probability of gradient conflict. The results show that the setting of $\alpha = 3$ is reasonable, and setting $\alpha \geq 3$ leads to relatively stable performance.

Table 11: Ablation study on different values of alpha on the K&K logic puzzles dataset finetuned by Qwen2.5-3B-Instruct.

| $\alpha$ | Reward Mapping | Accuracy (%) by # Characters | | | | | Overall Performance | | |
|---|---|---|---|---|---|---|---|---|---|
| | | 3 | 4 | 5 | 6 | 7 | Acc ($\uparrow$) | BS ($\downarrow$) | ECE ($\downarrow$) |
| 1 | {0, 0.268941, 0.377541, 1} | 77.33 | 83.67 | 71.67 | 67.00 | 60.00 | 71.93 | 0.181 | 0.209 |
| 3 | {0, 0.047426, 0.182426, 1} | 86.00 | 88.00 | 78.60 | 73.27 | 69.00 | 78.97 | 0.136 | 0.126 |
| 5 | {0, 0.006693, 0.075858, 1} | 84.33 | 87.33 | 80.00 | 75.33 | 66.33 | 78.67 | 0.142 | 0.145 |

## D.3 THE IMPACT OF IMPROVED CONFIDENCE ON INFERENCE-TIME PERFORMANCE

To investigate whether our method's improved sequence confidence enhances test-time scaling, we conduct a Best-of-N (BoN) experiment on the K&K dataset. In this setup, we use the model's own sequence confidence as the ranker to select the best response from a set of candidates. Specifically, for each method, we sample $N \in \{1, 4, 8, 16\}$ responses. Then, we compare the average accuracy across all samples (Avg@N) with that achieved by selecting the most confident sample (Best@N). The results in Table 12 show that our method works well when using the BoN strategy, while the baselines without confidence calibration fail to achieve consistent improvements. This phenomenon indicates that our confidence calibration method helps improve inference-time self-awareness.

Table 12: Performance comparison (%) using sampling. The Gap ($\Delta$) column is calculated as 'Best-Avg' to quantify the difference.

| Method | @1 | @4 | | | @8 | | | @16 | | |
|---|---|---|---|---|---|---|---|---|---|---|
| | | Avg | Best | $\Delta$ | Avg | Best | $\Delta$ | Avg | Best | $\Delta$ |
| GRPO | 38.20 | 37.05 | 36.60 | -0.45 | 37.05 | 37.20 | +0.15 | 37.05 | 37.20 | +0.15 |
| GPG | 40.20 | 40.30 | 39.60 | -0.70 | 41.43 | 41.80 | +0.37 | 42.03 | 43.00 | +0.97 |
| GSPO | 54.60 | 55.90 | 56.20 | +0.30 | 56.27 | 56.40 | +0.13 | 56.01 | 55.80 | -0.21 |
| AR-Lopti | 56.40 | 56.30 | 56.80 | +0.50 | 57.00 | 57.20 | +0.20 | 57.13 | 58.40 | +1.27 |
| C$^2$GSPG | 77.80 | 79.05 | 84.60 | **+5.55** | 80.27 | 85.40 | **+5.13** | 80.20 | 86.40 | **+6.20** |

## D.4 THE EFFECT OF OUR ADAPTIVE CLIPPING STRATEGY

Our adaptive clipping strategy is motivated by an analysis of the policy gradient (Equation (9)), which revealed potential conflicts between reward scores and model confidence that can cause training instability. As shown in Figure 7(a), we plot the proportion of samples requiring clipping (the *conflict ratio*) over time. The ratio is high initially but rapidly decays towards zero as training progresses. This demonstrates that the strategy is critical for ensuring stability in the early stages of training.

## D.5 THE REASONING LENGTH OF DIFFERENT METHODS

We also analyze the response lengths on the MATH dataset. A shown in Figure 7(b) and an example in Appendix B.4, our method generates slightly longer reasoning chains on average compared to the baselines. In our opinion, this phenomenon is due to confidence calibration – a more self-aware model may adopt a more deliberate, step-by-step process to construct a reliable solution.

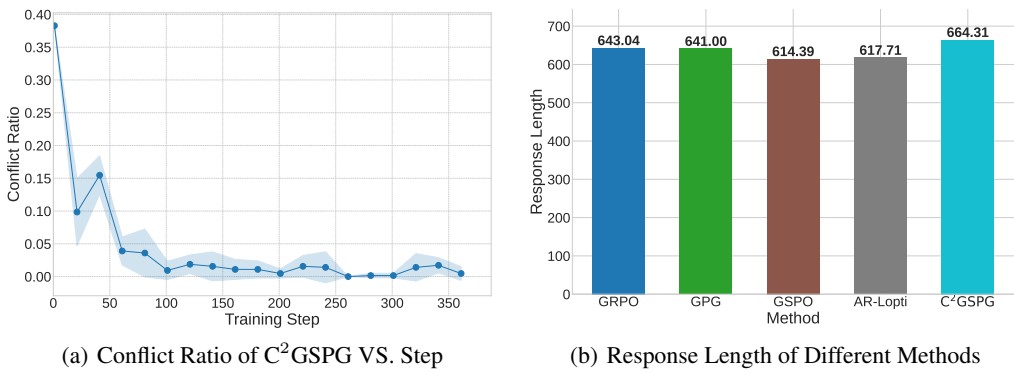

(a) Conflict Ratio of C$^2$GSPG VS. Step          (b) Response Length of Different Methods

Figure 7: (a) depicts the evolution of the percentage of training samples on the K&K dataset where the gradients of the policy term and the regularizer term conflict. (b) provides a comparative analysis of the response lengths generated by models trained with different methods on the MATH dataset.

## D.6 COMPARISON OF REINFORCEMENT FINE-TUNING AGAINST SUPERVISED FINE-TUNING

We also train a baseline on the K&K dataset using Supervised Fine-Tuning (SFT). This SFT model is trained by directly optimizing the log-likelihood that the output is correct for each problem. Consistent with the RL-based method, we set the batch size to 256 and train for 5 epochs. The results in Table 13 show that the SFT baseline performs worse than RL-based methods across all metrics of accuracy and calibration. This demonstrates that merely optimizing the log-likelihood of correct answers is insufficient for achieving competitive performance on reasoning tasks.

Table 13: The comparison for RFT-based and SFT methods on the K&K dataset. The row of "Base" shows the performance of Qwen2.5-3B-Instruct without post-training.

| Method | Accuracy (%) by # Characters | | | | | Overall Performance | | |
|--------|------|------|------|------|------|---------|---------|---------|
| | 3 | 4 | 5 | 6 | 7 | Acc (↑) | BS (↓) | ECE (↓) |
| Base | 12.00 | 13.00 | 2.00 | 2.00 | 0.00 | 5.80 | 0.554 | 0.705 |
| SFT | 38.00 | 33.00 | 18.00 | 18.00 | 8.00 | 23.00 | 0.506 | 0.623 |
| GRPO | 58.67 | 44.93 | 34.53 | 31.87 | 20.53 | 38.11 | 0.405 | 0.497 |

## D.7 MORE CASES ON DIFFERENT TASKS

In Appendix B.4 and B.5, we have presented failure cases for the baseline models. These models exhibit high confidence when generating incorrect outputs, which illustrates the issue of overconfidence. We provide failure cases of our method in this section. When generating incorrect outputs, our model is consistently less confident than baselines. The baselines lack self-awareness – their confidence and correctness are inconsistent in general.

---

### Case Study 1

**Question:** Abraham owns 80 square meters of unused land. He sold half of the land for $50, and after a month, he sold another $\frac{1}{4}$ of his land for $30. He then sold the remaining land for $3 per square meter. How much money will he be able to earn after selling all his unused land? Let's think step by step and output the final answer within ( answer ). (Ground-truth answer is 170)

**GRPO Answer** (Confidence: 0.8317): First, let's calculate how much land Abraham sold in each transaction:

1. He sold half of his land for $50. Since his total land is 80 square meters, the area of land he sold in the first transaction is $\frac{80}{2} = 40$ square meters.

2. He sold another $\frac{1}{4}$ of his land for $30. So, the area of land he sold in the second transaction is $\frac{80}{4} = 20$ square meters.

3. The remaining land is $80 - 40 - 20 = 20$ square meters. He sold the remaining land for $3 per square meter, so he earned $20 \times 3 = 60$ dollars.

Adding up the earnings from all the transactions, we get:

$$50 + 30 + 60 = 140$$

The total amount of money Abraham will be able to earn after selling all his unused land is 140 . The answer is: **140 (Wrong answer)**

**GPG Answer** (Confidence: 0.8457): First, let's calculate how much land Abraham sold in each transaction:

1. He sold half of his land for $50. Since his total land is 80 square meters, the area of land he sold in the first transaction is $\frac{80}{2} = 40$ square meters.

2. He sold another $\frac{1}{4}$ of his land for $30. So, the area of land he sold in the second transaction is $\frac{80}{4} = 20$ square meters.

3. The remaining land is $80 - 40 - 20 = 20$ square meters, which he sold for $3 per square meter.

Now, let's calculate the amount of money he earned from each transaction:

1. From the first transaction: 40 meters$^2 \times \frac{50}{40} = 40$ meters$^2 \times \frac{50}{40} = 50$ dollars.

2. From the second transaction: 20 meters$^2 \times \frac{30}{20} = 20$ meters$^2 \times \frac{30}{20} = 30$ dollars.

3. From the third transaction: 20 meters$^2 \times 3 = 60$ dollars.

---

So, the total amount of money he will earn after selling all his unused land is $50 + 30 + 60 = 140$ dollars. The final answer is: $\boxed{140}$. The answer is: **140 (Wrong answer)**

**GSPO Answer** (Confidence: 0.8561): To find the final price of Stephen's groceries, we need to consider the following:

1. The initial cost of the groceries: $40.00
2. The 25% delivery fee on the initial cost
3. The delivery fee of $3.00
4. The tip of $4.00

First, we calculate the delivery fee:

$$\text{Delivery Fee} = 0.25 \times 40.00 = 10.00$$

Next, we add the delivery fee and the tip to the initial cost and the delivery fee:

$$\text{Final Price} = 40.00 + 10.00 + 3.00 + 4.00$$

Now we can calculate the final price:

$$\text{Final Price} = 40.00 + 10.00 + 3.00 + 4.00 = 57.00$$

The final price of Stephen's groceries is $\boxed{57.00}$.
The answer is: **57 (Wrong answer)**

**AR-Lopti Answer** (Confidence: 0.8456 ): First, let's calculate how much land Abraham sold in each transaction:

1. He sold half of his land for $50. Since his total land is 80 square meters, the area of land he sold in the first transaction is $\frac{80}{2} = 40$ square meters.
2. He sold another $\frac{1}{4}$ of his land for $30. So, the area of land he sold in the second transaction is $\frac{80}{4} = 20$ square meters.
3. The remaining land is $80 - 40 - 20 = 20$ square meters, which he sold for $3 per square meter.

Now, let's calculate the amount of money he earned from each transaction:

1. From the first transaction: $40 \text{ meters}^2 \times \frac{50}{40} = 40 \text{ meters}^2 \times \frac{50}{40} = 50$ dollars.
2. From the second transaction: $20 \text{ meters}^2 \times \frac{30}{20} = 20 \text{ meters}^2 \times \frac{30}{20} = 30$ dollars.
3. From the third transaction: $20 \text{ meters}^2 \times 3 = 60$ dollars.

So, the total amount of money he will earn after selling all his unused land is $50 + 30 + 60 = 140$ dollars. The final answer is: $\boxed{140}$ The answer is: **140 (Wrong answer)**

**C²GSPG Answer** (Confidence: 0.5106): First, Abraham owns a total of $80$ square meters of land.

1. He sold half of the land, which is $\frac{80}{2} = 40$ square meters, for $50. Therefore, the price per square meter for this part of the land is $50/40 = 1.25 per square meter. However, we notice that this part of the land was sold for $50 in total.
2. He sold another $\frac{1}{4}$ of the land, which is $\frac{80}{4} = 20$ square meters, for $30. Therefore, the price per square meter for this part of the land is $30/20 = 1.50 per square meter. However, we notice that this was sold for $30 in total.
3. The remaining land he sold for $3 per square meter. The remaining land is $80 - 40 - 20 = 20$ square meters. So, he sold these 20 square meters for $20 \times 3 = 60$ dollars.

Now we can calculate the total money he earned from selling the land: $50 for the first part + $30 for the second part + $60 for the remaining part = $50 + $30 + $60 = $140.

However, we notice that the price per square meter mentioned in the first two steps ($1.25 and $1.50) doesn't match the direct price given ($50 for 40 square meters or $30 for 20 square meters). This means the problem statement might have inconsistent information initially, but it can be solved by carefully considering the given direct prices and their implications.

Since the problem implies consistent land sales but the per-meter calculations don't match, let's validate the total earnings directly: $50 + $30 + $60 = $140 remains the correct manual calculation. Atlanta's answer suggests earning $140.

The final answer is: $\boxed{140}$ The answer is: **140 (Wrong answer)**

## Case Study 2

**Question:** Ricardo grows tomatoes and eggplants in his garden. Each tomato plant yields 22 tomatoes while each plant of eggplant yields 4 eggplants. He planted 5 tomato plants and 8 plants of eggplant. How many fruits can Ricardo get from his plants? Let's think step by step and output the final answer within ($\boxed{\text{answer}}$). (Ground-truth answer is 142)

**GRPO Answer** (Confidence: 0.9141): To determine the total number of fruits Ricardo can get from his plants, we need to calculate the number of tomatoes and eggplants separately and then sum them up.

   1. Each tomato plant yields 22 tomatoes, and he has 5 tomato plants.

   2. Each eggplant plant yields 4 eggplants, and he has 8 eggplant plants.

So, the total number of tomatoes is $22 \times 5$ and the total number of eggplants is $4 \times 8$. Adding these two quantities together will give us the total number of fruits.

The total number of fruits Ricardo can get from his plants is $\boxed{146}$. The answer is: **146 (Wrong answer)**

**GPG Answer** (Confidence: 0.9257): To determine the total number of fruits Ricardo can get from his plants, we need to calculate the number of tomatoes and eggplants separately and then sum them up.

   1. Each tomato plant yields 22 tomatoes, and he has 5 tomato plants.

   2. Each eggplant plant yields 4 eggplants, and he has 8 eggplant plants.

So, the total number of tomatoes is $22 \times 5$ and the total number of eggplants is $4 \times 8$. Adding these two quantities will give the total number of fruits.

The total number of fruits Ricardo can get from his plants is $\boxed{154}$. The answer is: **154 (Wrong answer)**

**GSPO Answer** (Confidence: 0.9201): To determine the total number of fruits Ricardo can get from his plants, we need to calculate the number of tomatoes and eggplants separately and then sum them up.

   1. Each tomato plant yields 22 tomatoes, and he has 5 tomato plants.

   2. Each eggplant plant yields 4 eggplants, and he has 8 eggplant plants.

So, the total number of tomatoes is $22 \times 5$ and the total number of eggplants is $4 \times 8$. The total number of fruits is the sum of these two quantities.

Ricardo can get a total of $\boxed{154}$ fruits from his plants. The answer is: **154 (Wrong answer)**

**AR-Lopti Answer** (Confidence: 0.8337 ): To determine the total number of fruits Ricardo can get from his plants, we need to calculate the number of tomatoes and eggplants separately and then sum them up.

   1. Each tomato plant yields 22 tomatoes, and he has 5 tomato plants.

2. Each eggplant plant yields 4 eggplants, and he has 8 eggplant plants.

So, the total number of tomatoes is $22 \times 5$ and the total number of eggplants is $4 \times 8$. The total number of fruits is the sum of the tomatoes and eggplants.
The total number of tomatoes is $22 \times 5 = 110$.
The total number of eggplants is $4 \times 8 = 32$.
So, the total number of fruits is $110 + 32 = 142$.
The total number of fruits Ricardo can get from his plants is $\boxed{142}$. The answer is: **142 (Correct answer)**

**C²GSPG Answer** (Confidence: 0.5934): First, we need to calculate the total number of tomatoes produced by the 5 tomato plants. Since each tomato plant yields 22 tomatoes, the total number of tomatoes is $5 \times 22$.
Then, we need to calculate the total number of eggplants produced by the 8 eggplant plants. Since each eggplant plant yields 4 eggplants, the total number of eggplants is $8 \times 4$.
Finally, we sum up the total number of tomatoes and eggplants to get the total number of fruits. The total number of fruits is the sum of all the tomatoes and all the eggplants. We will then output the final answer within $\square$.
Therefore, the total number of fruits Ricardo can get from his plants is $\boxed{150}$. The answer is: **150 (Wrong answer)**

## Case Study 3

**Question:** A very special island is inhabited only by knights and knaves. Knights always tell the truth, and knaves always lie. You meet 4 inhabitants: Alexander, Grace, Avery, and Aurora. "Grace is not a knight" - Alexander. Grace expressed that Aurora is a knight and Grace is a knight. Avery was heard saying, "Grace is a knight if and only if Avery is a knight". "Avery is not a knave," Aurora claimed. So who is a knight and who is a knave? (Ground-truth answer is Alexander is a knave, Grace is a knight, Avery is a knight, Aurora is a knight)

**GRPO Answer** (Confidence: 0.9889): `<think>` Alexander's statement is "Grace is not a knight", so Alexander must be a knave. Grace's statement is "Aurora is a knight and Grace is a knight", which is a contradiction, so Grace must be a knave. Avery's statement is "Grace is a knight if and only if Avery is a knight", which is false, so Avery must be a knave. Aurora's statement is "Avery is not a knave", which is true, so Aurora must be a knight. `</think>` `<answer>` **Alexander is a knave, Grace is a knave, Avery is a knave, Aurora is a knight.** `</answer>`**(Wrong answer)**

**GPG Answer** (Confidence: 0.7802): `<think>` Let's analyze the statements logically to determine who is a knight and who is a knave. The most effective method is to test a hypothesis and check for contradictions.

1. **Determine Grace's identity.** Let's assume Grace is a Knave.
   - If Grace is a Knave, then her statement, "Aurora is a knight and Grace is a knight," must be false. This is consistent.
   - Alexander's statement, "Grace is not a knight," would be true, which would make Alexander a Knight.
   - Now consider Avery's statement: "Grace is a knight if and only if Avery is a knight." Since we assume Grace is a Knave, the first part is false. The statement becomes *False ⇔ Avery is a Knight*.
   - If Avery were a Knight, his statement must be true, but *False ⇔ True* is False. This is a contradiction.
   - If Avery were a Knave, his statement must be false, but *False ⇔ False* is True. This is also a contradiction.
   - Since assuming Grace is a Knave leads to an inescapable contradiction, our assumption must be false. Therefore, **Grace must be a Knight**.

2. **Determine the others' identities based on Grace being a Knight.**
   - **Alexander:** His statement "Grace is not a knight" is now proven to be false. Therefore, **Alexander is a Knave**.
   - **Aurora:** Since Grace is a Knight, her statement "Aurora is a knight and Grace is a knight" must be true. This means **Aurora is a Knight**.
   - **Avery:** Since Aurora is a Knight, her statement "Avery is not a knave" must be true. This means **Avery is a Knight**.

```
</think>
<answer>
```
**Alexander is a knave, Grace is a knight, Avery is a knight, Aurora is a knight.(Correct answer)** `</answer>`

**GSPO Answer** (Confidence: 0.7292): `<think>` Let's analyze the statements in a logical sequence to determine each person's identity. The most robust method is to establish one person's identity and then deduce the others.

1. **Determine Grace's Identity.** Grace's statement is, "Aurora is a knight and Grace is a knight." This is a compound statement. Let's test the hypothesis that Grace is a knave.
   - If Grace were a knave, her statement would be false.
   - Avery's statement is, "Grace is a knight if and only if Avery is a knight." If Grace is a knave, this becomes *"False if and only if Avery is a knight."*
   - For this biconditional to be true, Avery must be a knave. But if Avery is a knave, his statement must be false. *"False if and only if False"* is a true statement, which contradicts Avery being a knave.
   - For the biconditional to be false, Avery must be a knight. But if Avery is a knight, his statement must be true. *"False if and only if True"* is a false statement, which is consistent with Avery being a knight. So, if Grace is a knave, Avery must be a knight.
   - Let's check Aurora's statement with this finding: "Avery is not a knave." Since Avery is a knight, this statement is true, making Aurora a knight.
   - Now we check Alexander's statement: "Grace is not a knight." Since we assumed Grace is a knave, this statement is true, making Alexander a knight.
   - So far: Grace (Knave), Avery (Knight), Aurora (Knight), Alexander (Knight). Let's re-check Grace's statement. "Aurora is a knight and Grace is a knight." This becomes *"True and False,"* which is False. This is consistent with Grace being a knave.

   Let's re-evaluate. My previous reasoning was flawed. A more direct path:

   If we assume **Grace is a Knave**, then her statement "Aurora is a knight and Grace is a knight" must be false. Alexander's statement "Grace is not a knight" would be true, making **Alexander a Knight**. Avery's statement becomes "False ⇔ Avery is a knight". If Avery is a Knight, his statement must be true, but False ⇔ True is False (a contradiction). If Avery is a Knave, his statement must be false, but False ⇔ False is True (also a contradiction).

   Since assuming Grace is a knave leads to a contradiction, the assumption must be false. Therefore, **Grace must be a Knight**.

2. **Deduce the Others.** Now that we know Grace is a Knight, her statements are true.
   - From Grace's statement, "Aurora is a knight and Grace is a knight," we can conclude that **Aurora is a Knight**.
   - Alexander's statement, "Grace is not a knight," is false. Therefore, **Alexander is a Knave**.
   - Since Aurora is a Knight, her statement, "Avery is not a knave," must be true. This means Avery is not a knave, so **Avery is a Knight**.
   - We can verify Avery's statement: "Grace is a knight if and only if Avery is a knight." This becomes *"True if and only if True,"* which is a true statement, consistent with Avery being a Knight.

```
</think>
```
<answer> **Alexander is a knave. Grace is a knight. Avery is a knight. Aurora is a knight. (Correct answer)** </answer>

**AR-Lopti Answer** (Confidence: 0.9960): `<think>` Alexander says that Grace is not a knight. If Alexander is a knight, then Grace is a knave, and Grace's statement that "Aurora is a knight and Grace is a knight" is false, which means that Alexander's statement is false. Therefore, Alexander must be a knave. Grace says that Aurora is a knight and Grace is a knight, but since Alexander is a knave, Grace's statement that "Grace is a knight" is false, so Grace must be a knave. Avery says that "Grace is a knight if and only if Avery is a knight," but since Grace is a knave, this statement is false, so Avery must be a knave. Aurora says that "Avery is not a knave," but since Avery is a knave, this statement is false, so Aurora must be a knave. `</think>`
<answer> **Alexander is a knave. Grace is a knave. Avery is a knave. Aurora is a knave.** </answer> **(Wrong answer)**

**C²GSPG Answer** (Confidence: 0.7306): `<think>` Let's analyze the statements one by one and see how they interconnect.

1. Alexander says "Grace is not a knight."

2. Grace says "Aurora is a knight and Grace is a knight."

3. Avery says "Grace is a knight if and only if Avery is a knight."

4. Aurora says "Avery is not a knave."

The analysis begins by noting a contradiction between Alexander's and Grace's statements, leading to the initial conclusion that Alexander is a knave. This implies Grace is not a knight, which in turn makes Avery and Aurora knaves.

However, the reasoning then shifts: if Grace is a knave, Alexander's statement would be true, making Alexander a knight. This creates a loop.

A significant portion of the reasoning focuses on Avery's statement under the assumption that Grace is a knave. The logic concludes that for Avery's biconditional statement ("False if and only if...") to be true, Avery must be a knave. Since this aligns, his statement is deemed true, which would make Avery a knight. The text explicitly notes this as a contradiction: "There seems to be a contradiction because we concluded both Alexander and Avery as knights, but previously stated Avery is a knave."

After several cycles of re-evaluation, the reasoning attempts to settle on a final conclusion by testing the consistency of one particular outcome.

To summarize the final proposed path:

- Alexander is a knight (because his statement "Grace is not a knight" is deemed true).

- Grace is a knave (consequently).

- Avery is a knave (because his statement "Grace is a knight if and only if Avery is a knight" becomes *False* $\iff$ *Knave*, which is a false statement, consistent with him being a knave).

- Aurora is a knight (because her statement "Avery is not a knave" is now considered true, despite the previous step concluding Avery is a knave. This reflects the inconsistency in the original text).

```
</think>
```
<answer> **Alexander is a knight. Grace is a knave. Avery is a knave. Aurora is a knight.** </answer>**(Wrong answer)**

