# OpenReview forum: "C$^2$GSPG: Confidence-calibrated Group Sequence Policy Gradient towards Self-aware Reasoning"
_ICLR.cc/2026/Conference — Submitted to ICLR 2026_

### Official Review · Reviewer_s7fz · 2025-11-01

**Soundness:** 4
**Presentation:** 4
**Contribution:** 4
**Rating:** 8
**Confidence:** 3

**Summary:**

The paper presents C$^2$GSPG (Confidence-Calibrated Group Sequence Policy Gradient), a reinforcement learning framework designed to alleviate the overconfidence problem observed in GRPO-style reasoning models.
By introducing a sequence-level confidence measure and a binary cross-entropy regularizer that aligns model confidence with reward, the method encourages self-aware reasoning—increasing confidence for correct generations and reducing it for incorrect ones.
The paper also proposes sigmoid-based reward normalization and adaptive clipping to handle non-binary reward settings.
Extensive experiments on logical and mathematical reasoning benchmarks show consistent improvements in both accuracy and calibration metrics (ECE, Brier Score).

**Strengths:**

1. Clear and well-written: The paper is easy to follow with solid organization and a strong motivation.

2. Reasonable and well-motivated algorithmic design: Each component (confidence-modulated advantage, BCE regularizer, adaptive clipping) is conceptually sound and technically justified.

3. The work tackles an important problem of reasoning LLMs — self-awareness and confidence calibration, which is a key limitation of current LLMs.

4. Empirical results show consistent and significant performance improvements, not only in reasoning accuracy but also in calibration robustness across multiple datasets.

5. The method effectively generalizes from binary to non-binary reward settings, demonstrating broad applicability.

6. Calibration metrics (Brier Score, ECE) are appropriately chosen and clearly analyzed, showing that confidence calibration contributes directly to better reasoning quality.

**Weaknesses:**

1. The paper lacks fine-grained ablation studies to isolate the contribution of each component (e.g., GSPG, BCE, adaptive clipping).

2. Experiments are limited to small-sized backbones (1.5B–3B); validation on larger LLMs would strengthen the claim of robustness and scalability.

3. It remains unclear whether the calibrated confidence translates into improved inference-time self-awareness (e.g., via self-verification or test-time scaling strategies).

4. The method involves several heuristic hyperparameters (α, β, clipping threshold) whose tuning process or sensitivity is not well discussed.

**Questions:**

1. Could the improved sequence confidence be used to enhance test-time scaling (e.g., weighted majority voting or Best-of-N)?

2. Can the authors provide component-level ablations or at least a discussion on expected effects when removing each module (e.g., normalization, clipping, or BCE)?

3. The qualitative examples mostly show correct generations—could you also include failure cases and their confidence scores to illustrate calibration behavior for incorrect outputs?

4. Have you observed whether confidence calibration affects generation dynamics, such as reasoning length or exploration behavior?

---

> ### Author Response · Authors · 2025-11-20
> **Response to Reviewer s7fz**
>
> Thank you for your appreciation of our work. Below, we resolve your concerns one by one. All required experiments are included in our revised paper, and some are shown below for quick review.
>
> **Q1: Add more ablation studies**
>
> **A1:** Fine-grained ablation studies are provided:
>
> 1) For our BCE regularizer and GSPG framework, we design an ablation study on K\&K using Qwen2.5-3B-Instruct. For each method, we augment it with our BCE regularizer. **Our regularizer improves all the methods, and GSPG outperforms all the baselines (Table 9 in Appendix D.2).**
>
> ||3|4|5|6|7|Acc|BS|ECE|
> |-|-|-|-|-|-|-|-|-|
> |GRPO|58.67|44.93|34.53|31.87|20.53|38.11|0.405|0.497|
> |GRPO+BCE|66.60|57.07|45.80|41.13|29.73|48.07|0.335|0.409|
> |GPG|69.53|64.80|49.47|43.20|35.93|52.59|0.363|0.363|
> |GPG+BCE|70.93|66.07|53.07|43.87|36.33|54.05|0.337|0.330|
> |GSPO|71.13|66.40|55.40|48.20|40.07|56.24|0.291|0.302|
> |GSPO+BCE|77.07|76.53|61.80|55.73|47.27|63.68|0.231|0.233|
> |AR-Lopti|73.20|64.93|54.13|49.47|41.33|56.61|0.282|0.343|
> |AR-Lopti+BCE|78.40|81.80|70.27|66.13|58.93|71.11|0.192|0.225|
> |GSPG|79.50|75.50|67.00|57.00|50.00|65.80|0.215|0.256|
> |GSPG+BCE(C²GSPG)|**86.00**|**88.00**|**78.60**|**73.27**|**69.00**|**78.97**|**0.136**|**0.126**|
>
> 2) Adaptive clipping: Its usefulness has been shown by Table 4, with analysis in Lines 415–424.
>
> 3) Reward normalization: For a non-binary reward not in [0, 1], the normalization is necessary to make the BCE regularizer valid. See Lines 265-271
>
> **Q2: Test on larger LLMs**
>
> **A2:** Following AR-Lopti, we fine-tune Qwen2.5-7B for math reasoning and Qwen2.5-7B-Instruct-1M for logic reasoning. **Our method outperforms baselines on larger LLMs (Tables 7 and 8 in Appendix D.1), demonstrating its scalability.**
>
> ||GSM8K|MATH500|AMC23|AIME24|Olympiad|Minerva|Acc|BS|ECE|
> |-|-|-|-|-|-|-|-|-|-|
> |GRPO|89.60|72.04|43.87|16.33|**25.10**|23.82|45.13|0.531|0.535|
> |GPG|89.63|**73.24**|51.56|13.00|24.62|23.75|45.97|0.515|0.520|
> |GSPO|88.75|72.44|**53.19**|14.67|24.56|22.35|45.99|0.520|0.524|
> |AR-Lopti|88.72|73.08|47.31|14.42|24.65|**24.26**|45.41|0.522|0.528|
> |C²GSPG|**90.60**|71.56|52.12|**20.67**|24.44|22.94|**47.06**|**0.511**|**0.515**|
>
> ||3|4|5|6|7|Acc|BS|ECE|
> |-|-|-|-|-|-|-|-|-|
> |GRPO|79.00|72.40|56.40|56.60|42.40|61.36|0.254|0.308|
> |GPG|90.60|88.20|73.20|69.60|53.60|75.04|0.190|0.194|
> |GSPO|84.00|88.00|75.20|72.00|67.20|77.28|0.157|0.171|
> |AR-Lopti|82.60|87.60|71.60|68.00|58.40|73.64|0.180|0.214|
> |C²GSPG|**95.60**|**92.80**|**85.80**|**77.00**|**73.00**|**84.84**|**0.126**|**0.124**|
>
> **Q3: Usefulness in test-time scaling**
>
> **A3:** We conduct a Best-of-N experiment on K\&K. For each model, we sample $N\in\{4,8,16\}$ responses and compare the average accuracy (Avg@N) with that of the most confident response (Best@N). **Confidence calibration improves test-time self-awareness --- using the BoN strategy, our method works well, while the baselines fail to achieve consistent improvements (Table 12 in Appendix D.3).**
>
> ||Avg@4|Best@4|$\Delta$@4|Avg@8|Best@8|$\Delta$@8|Avg@16|Best@16|$\Delta$@16|
> |-|-|-|-|-|-|-|-|-|-|
> |GRPO|37.05|36.60|-0.45|37.05|37.20|+0.15|37.05|37.20|+0.15|
> |GPG|40.30|39.60|-0.70|41.43|41.80|+0.37|42.03|43.00|+0.97|
> |GSPO|55.90|56.20|+0.30|56.27|56.40|+0.13|56.01|55.80|-0.21|
> |AR-Lopti|56.30|56.80|+0.50|57.00|57.20|+0.20|57.13|58.40|+1.27|
> |C²GSPG|79.05|84.60|**+5.55**|80.27|85.40|**+5.13**|80.20|86.40|**+6.20**|
>
> **Q4: Robustness**
>
> **A4:** We fine-tune Qwen2.5-3B-Instruct on K\&K under different settings.
>
> 1) The transformed rewards lose their distinguishability when $\alpha$ is too large and cause gradient conflict when it is too small. Setting $\alpha$ in [3, 5] leads to stable performance (Table 11 in Appendix D.2).
>
> |$\alpha$|3|4|5|6|7|Acc|BS|ECE|
> |-|-|-|-|-|-|-|-|-|
> |1|77.33|83.67|71.67|67.00|60.00|71.93|0.181|0.209|
> |3|86.00|88.00|78.60|73.27|69.00|78.97|0.136|0.126|
> |5|84.33|87.33|80.00|75.33|66.33|78.67|0.142|0.145|
>
> 2) Setting $\beta=0.03$ leads to the best performance. Notably, the models using other settings (0.01 or 0.05) still outperform the baselines.
>
> |$\beta$|3|4|5|6|7|Acc|BS|ECE|
> |-|-|-|-|-|-|-|-|-|
> |0.01|82.00|83.00|70.00|64.33|61.00|72.07|0.169|0.171|
> |0.03|86.00|88.00|78.60|73.27|69.00|78.97|0.136|0.126|
> |0.05|81.00|83.00|73.00|65.33|58.33|72.13|0.168|0.132|
>
> 3) We used an adaptive, rather than fixed, clipping threshold. See Lines 277-279.
>
> **Q5: More failure cases**
>
> **A5:** In Appendices B.4 & B.5, we have shown failure cases for the baseline models, clearly indicating their overconfidence. We present failure cases for our method in Appendix D.7, where our model is self-aware and has lower confidence than the baselines.
>
> **Q6: The impact on response length**
>
> **A6:** Our method generates longer responses on average (Figure 7(b) in Appendix D.5), implying that a confidence-calibrated model constructs solutions deliberately.
>
> We hope the above response resolves your concerns and makes you more confident in supporting our work. Thanks in advance for your consideration.

---

### Official Review · Reviewer_QD6C · 2025-11-01

**Soundness:** 4
**Presentation:** 3
**Contribution:** 4
**Rating:** 6
**Confidence:** 3

**Summary:**

This paper introduces Confidence-Calibrated Group Sequence Policy Gradient, a method designed to address the overconfidence issue in LLMs for reasoning tasks. By incorporating a confidence calibration regularizer into the GRPO framework, C2GSPG aligns model confidence with actual correctness, reducing overconfidence in incorrect answers. The method introduces a Brier Score regularizer to ensure confidence estimates are reliable, improving both reasoning accuracy and calibration. Experimental results demonstrate that C^2GSPG outperforms sota methods in accuracy and confidence calibration, particularly on complex tasks with higher difficulty levels.

**Strengths:**

1. The paper introduces C2GSPG, a novel method that uses confidence calibration to mitigate the overconfidence problem in LLMs

2.It shows a large improvement in logical reasoning tasks, outperforming state-of-the-art methods like AR-Lopti and GRPO, demonstrating its effectiveness in challenging reasoning scenarios.

3. The method not only enhances accuracy but also significantly improves confidence calibration.

**Weaknesses:**

1. I am confused about the difference between aligning confidence (probability) with reward versus directly aligning output with reward. The paper also doesn't explain the practical consequences of the overconfidence problem. I'm not sure how exactly does overconfidence affect performance in real-world tasks?

2. C^2GSPG performs well on the K&K dataset, but the improvements in mathematical reasoning are not significant, especially in Brier Score. More substantial results in this area would strengthen the method’s broader applicability.

3. The innovation of adding Brier Score to the GSPO objective seems limited, as it is already a widely used method for confidence calibration.

**Questions:**

1. Does C^2GSPG scale well on larger models?

2. Can you provide a concrete example of a scenario where the overconfidence problem leads to poor model performance?

---

> ### Author Response · Authors · 2025-11-20
> **Response to Reviewer QD6C**
>
> Thank you for your positive feedback on our work. Below, we answer your questions one by one. All required experiments are included in our revised paper, and some are shown below for quick review.
>
> **Q1: The reason for aligning confidence, rather than output, to reward. Provide an example for its rationality**
>
> **A1:** 'Directly aligning output with reward' means maximizing token-level output correctness, which is exactly what the current methods (e.g., SFT and GRPO) do. However, these approaches have been shown to incentivize overconfident guessing rather than genuine self-awareness [1-3]. **Our confidence defined in Eq.(5) is the average token-level likelihood of the sequence. It reflects sequence-level certainty rather than token-level probability of the answer.** Aligning the confidence with the reward helps models be aware of their potential mistakes.
>
> The examples in Appendices B.4 & B.5 illustrate the overconfidence issue: without confidence calibration, the baselines assign very high confidence to incorrect answers, which can easily mislead users.
>
> The negative impacts of overconfident models have been well-studied [4-5]. In risk-sensitive applications such as disease diagnosis and auto-driving, confidence scores are key signals for human auditing or fallback mechanisms. Overconfident models are unreliable because their confidence cannot reflect their mistake risk.
>
> We have added the above content to the revised paper.
>
> [1] Taming Overconfidence in LLMs: Reward Calibration in RLHF, 2024
>
> [2] Uncalibrated Reasoning: GRPO Induces Overconfidence for Stochastic Outcomes, 2025
>
> [3] Rewarding Doubt: A Reinforcement Learning Approach to Calibrated Confidence Expression of Large Language Models, 2025
>
> [4] Large Language Models lack essential metacognition for reliable medical reasoning, 2025
>
> [5] Reducing Overconfidence Predictions in Autonomous Driving Perception, 2022
>
> **Q2: Test on larger LLMs**
>
> **A2:** Following AR-Lopti, we fine-tune Qwen2.5-7B for math reasoning and Qwen2.5-7B-Instruct-1M for logic reasoning. **Our method outperforms baselines on larger LLMs, demonstrating its scalability. (Tables 7 and 8 in Appendix D.1)**
>
> ||GSM8K|MATH500|AMC23|AIME24|Olympiad|Minerva|Acc|BS|ECE|
> |-|-|-|-|-|-|-|-|-|-|
> |GRPO|89.60|72.04|43.87|16.33|**25.10**|23.82|45.13|0.531|0.535|
> |GPG|89.63|**73.24**|51.56|13.00|24.62|23.75|45.97|0.515|0.520|
> |GSPO|88.75|72.44|**53.19**|14.67|24.56|22.35|45.99|0.520|0.524|
> |AR-Lopti|88.72|73.08|47.31|14.42|24.65|**24.26**|45.41|0.522|0.528|
> |C²GSPG|**90.60**|71.56|52.12|**20.67**|24.44|22.94|**47.06**|**0.511**|**0.515**|
>
> ||3|4|5|6|7|Acc|BS|ECE|
> |-|-|-|-|-|-|-|-|-|
> |GRPO|79.00|72.40|56.40|56.60|42.40|61.36|0.254|0.308|
> |GPG|90.60|88.20|73.20|69.60|53.60|75.04|0.190|0.194|
> |GSPO|84.00|88.00|75.20|72.00|67.20|77.28|0.157|0.171|
> |AR-Lopti|82.60|87.60|71.60|68.00|58.40|73.64|0.180|0.214|
> |C²GSPG|**95.60**|**92.80**|**85.80**|**77.00**|**73.00**|**84.84**|**0.126**|**0.124**|
>
> **Q3: Analyze the results**
>
> **A3:** Our method performs well on K&K but incrementally in math reasoning. **This phenomenon is common [5-7] due to the nature of these two datasets.**
>
> 1. K&K is an unseen but simple task. It is based on the Knights and Knaves reasoning setting [1], with rule-based rewards [2]. K&K does not appear in the pre-training corpora of mainstream LLMs, making it an excellent dataset for evaluating RL methods [3-7] -- performance improvements primarily stem from learning rather than data memorization.
>
> 2. MATH is a seen but difficult task. Although LLMs were trained on math content, their capabilities are still insufficient to solve MATH problems. Current RL methods, including ours, cannot exceed the upper bound of these LLMs' capabilities.
>
> [1] On memorization of large language models in logical reasoning, 2024
>
> [2] Logic-rl: Unleashing LLM reasoning with rule-based reinforcement learning, 2025
>
> [3] REINFORCE++: Stabilizing Critic-Free Policy Optimization with Global Normalization, 2025
>
> [4] DUMP: Automated Distribution-Level Curriculum Learning for RL-based LLM Post-training, 2025
>
> [5] A Simple "Motivation" Can Enhance Reinforcement Finetuning of Large Reasoning Models, 2025
>
> [6] Do Not Let Low-Probability Tokens Over-Dominate in RL for LLMs, 2025
>
> [7] Efficient RL Training for Reasoning Models via Length-Aware Optimization, 2025
>
> **Q3: The innovation of adding the Brier Score to the GSPO objective seems limited**
>
> **A3:** We believe it is a misunderstanding: **Our method does NOT add the Brier Score (BS) into the GSPO objective.** Lines 184–190 and 274–279 show our objective: **GSPG with BCE regularization**. The Brier Score is solely an evaluation metric (Lines 353-355).
>
> We hope the above response resolves your concerns and eliminates any misunderstandings, making you more confident in raising your score. Thanks in advance for your consideration.

---

### Official Review · Reviewer_CHKD · 2025-11-01

**Soundness:** 2
**Presentation:** 2
**Contribution:** 2
**Rating:** 4
**Confidence:** 4

**Summary:**

The authors propose $C^{2}GSPG$, a novel confidence-calibrated group sequence policy gradient method. The core idea is twofold: first, it proposes a Group Sequence Policy Gradient (GSPG) framework that operates at the sequence level, aiming to eliminate the token-level biases found in GRPO. Second, it introduces a binary cross-entropy (BCE) regularizer to explicitly align the model's confidence (defined as the normalized sequence-level probability) with the actual reward of the sequence.

**Strengths:**

1. The method demonstrates good performance outperforming baselines.
2. The model can achieve some calibration on confidence and accuracy. This might be useful to improve interpretability.

**Weaknesses:**

1. I don't get it why we need to care about the model being overconfident if accuracy is good. An extreme example would be that a model that is 100% accurate and 100% overconfident is a perfect model. The paper fails to provide a compelling case for why we should sacrifice any accuracy for better calibration, which is a trade-off demonstrated by the $\beta$ hyperparameter in Table 5. The objective seems to be optimizing the confidence calibratio at the risk of distracting from accuracy.
2. The paper's GSPG framework is presented as a key contribution for eliminating token-level bias. However, this is very similar in principle to other recent sequence-level methods like GSPO. Both GSPG and GSPO compute a sequence-level advantage and apply a gradient based on the entire sequence probability. The introduction of sigmoid reward normalization and, in particular, the adaptive regularizer clipping, which simply turns off the regularizer if its gradient conflicts with the policy gradient, seems like an engineering patch to force the method to work rather than a principled solution. This contribution appears to be marginal at best.
3. The extension to non-binary rewards feels heuristic and ad-hoc. It is somewhat problematic since the authors do observe gradient conflict in ther empirical studies.

Overall, I think the paper has many flaws despite its good performace.

**Questions:**

See the above weaknesses.

---

> ### Author Response · Authors · 2025-11-20
> **Response to Reviewer CHKD (Part 1)**
>
> Below, we answer your questions one by one. All required experiments are included in our revised paper, and some are shown below for quick review.
>
> **Q1: Why is overconfidence an important problem?**
>
> **A1:** The short answer is **an overconfident model cannot be 100% accurate, in both theory and practice**.
>
> Overconfidence is a well-known, long-standing, and critical issue in machine learning [1,2,3]. The problem is first studied in the context of classification problems [4,5,6] -- a well-trained classifier may make incorrect predictions with high label probabilities (i.e., be overconfident). In reinforcement learning, this issue has also been found and discussed in many works [7,8,9]. This reduces the model's trustworthiness and may lead to significant failures in downstream applications. In risk-sensitive AI applications such as disease diagnosis and auto-driving, confidence scores are key signals for introducing human auditing or fallback mechanisms. In such situations, overconfident models are unreliable and thus inapplicable because their confidence cannot reflect their error risk.
>
> Essentially, the overconfidence stems from the model being forced to fit only the training data. Such a single objective incentivizes overconfident guessing rather than genuine self-awareness. **In summary, an overconfident model is always undesirable, and sometimes dangerous, making confidence calibration an important topic in the community.**
>
> [1] On Calibration of Modern Neural Networks, 2017
>
> [2] Predicting good probabilities with supervised learning, 2005
>
> [3] Explaining and harnessing adversarial examples, 2014
>
> [4] Deep Neural Networks are Easily Fooled: High Confidence Predictions for Unrecognizable Images, 2015
>
> [5] Evaluating model calibration in classification, 2019
>
> [6] Transforming Classifier Scores into Accurate Multiclass Probability Estimates, 2002
>
> [7] CoDaPo: Confidence and difficulty-adaptive policy optimization for post-training language models, 2025
>
> [8] Mitigating overconfidence in large language models: A behavioral lens on confidence estimation and calibration, 2025
>
> [9] Beyond Binary Rewards: Training LMs to Reason About Their Uncertainty, 2025
>
> **Q2: Does confidence calibration sacrifice accuracy?**
>
> **A2:** This is a good question. In Table 5 (math reasoning), overly strong confidence calibration (with a large $\beta$) leads to slight accuracy degradation while improving BS and ECE. However, in Table 4 (logic reasoning on K&K), we see that confidence calibration leads to significant improvements across all metrics. **In our opinion, this inconsistency stems from the nature of the two tasks in the context of current LLMs:**
>
> 1. K&K is an unseen but relatively simple task. It is based on the Knights and Knaves reasoning setting [1], with rule-based rewards [2]. K&K does not appear in the pre-training corpora of mainstream LLMs, making it an excellent dataset for evaluating RL methods [3-6] -- performance improvements primarily stem from learning rather than data memorization.
>
> 2. MATH is a seen but difficult task. Although LLMs were trained on math content, their capabilities are still insufficient to solve the problems in the MATH datasets (which contain a limited number of hard questions). Current RL methods, including ours, cannot exceed the upper bound of these LLMs' capabilities.
>
> From this viewpoint, each RL method's performance on the current MATH datasets is influenced by various factors (e.g., data size, model capability, etc.). As a result, the results achieved on K&K are more reliable for evaluating the rationality and effectiveness of the RL method itself.
>
> **In Proposition 3.1, we have demonstrated that the proposed confidence calibration term shares the same gradient direction as the GSPG loss.** It implies that, in theory, the weight of the confidence calibration ($\beta$) controls the step size/learning rate during training when the reward is binary.
>
> **In summary, the confidence calibration does not sacrifice accuracy in theory, and most of the experimental results in our work have provided positive evidence. However, when we use a model with insufficient capability to handle a challenging task (e.g., MATH), adding additional regularizers always further restricts its capability and harms model performance. It is natural, not a drawback of our method.**
>
> [1] On memorization of large language models in logical reasoning, 2024
>
> [2] Logic-rl: Unleashing LLM reasoning with rule-based reinforcement learning, 2025
>
> [3] REINFORCE++: Stabilizing Critic-Free Policy Optimization with Global Normalization, 2025
>
> [4] DUMP: Automated Distribution-Level Curriculum Learning for RL-based LLM Post-training, 2025
>
> [5] A Simple "Motivation" Can Enhance Reinforcement Finetuning of Large Reasoning Models, 2025
>
> [6] Efficient RL Training for Reasoning Models via Length-Aware Optimization, 2025

---

> ### Author Response · Authors · 2025-11-20
> **Response to Reviewer CHKD (Part 2)**
>
> **Q3: The robustness of our method to $\beta$.**
>
> **A3:** Besides using various $\beta$'s in math reasoning tasks, We fine-tune Qwen2.5-3B-Instruct on K&K with $\beta\in\{0.01, 0.03, 0.05\}$ to further study its impact. Table 10 in Appendix D.2 shows that:
>
> 1) Setting $\beta=0.03$  leads to the best performance.
>
> 2) Notably, the models using other settings (0.01 or 0.05) still outperform all the baseline methods.
>
> 3) All three metrics (Acc, BS, and ECE) change consistently. This phenomenon verifies our claim in Proposition 3.1 and provides new and strong evidence that confidence calibration does not harm model performance..
>
> |$\beta$|3|4|5|6|7|Acc|BS|ECE|
> |:-:|:-:|:-:|:-:|:-:|:-:|:-:|:-:|:-:|
> |0.01|82.00|83.00|70.00|64.33|61.00|72.07|0.169|0.171|
> |0.03|86.00|88.00|78.60|73.27|69.00|78.97|0.136|0.126|
> |0.05|81.00|83.00|73.00|65.33|58.33|72.13|0.168|0.132|
>
> In summary, these results demonstrate the usefulness of our calibration term and its robustness to $\beta$.
>
> **Q4: The novelty of GSPG**
>
> **A4:** For the GSPG framework used in our C²GSPG method, its differences compared to GSPO include two key points:
>
> 1) **Advantage formulations**: GSPO's advantage (Lines 106-107) depends purely on reward statistics, while GSPG's (Lines 188-190) is explicitly conditioned on sequence-level model confidence.
>
> 2) **Mitigation of sequence-level bias:** The objective function of GSPO (Lines 128-131) aims to eliminate token-level bias for stability, which can also be achieved by our GSPG framework (See Lines 241-251). However, as shown in Table 2, GSPO exhibits a bias towards high-confidence sequences, making it susceptible to overconfidence. C²GSPG avoids this issue by leveraging the unique advantage design to regulate gradient scaling, ensuring better calibration. See Lines 252-260 for more details.
>
> **Q6: The rationality of sigmoid reward normalization and adaptive regularizer clipping**
>
> **A6:** **Such extensions to non-binary rewards are motivated by our theoretical analysis, NOT heuristic.**
>
> 1) In principle, using sigmoid normalization is motivated by Proposition 3.1 — it pushes non-binary rewards to 0 or 1, reducing the probability of gradient conflicts by making the transformed rewards more closely match binary rewards. In addition, for a non-binary reward whose range is not from 0 to 1, our normalization strategy is necessary to make our BCE regularizer valid in practice.
>
> 2) For the remaining sequences suffering gradient conflicts, we use the adaptive clipping strategy to avoid the conflicts. Note that, in Figure 7(b) in Appendix D.6, we plot the proportion of samples requiring clipping (the 'conflict ratio') over time. The ratio is initially high but rapidly decays to zero as training progresses. This demonstrates that the strategy is critical for ensuring stability in the early stages of training.
>
> **The above two modules are friendly to engineering implementation, which should be considered an advantage rather than a weakness of our method:** Many state-of-the-art RL methods (such as GRPO) rely on substantial engineering design choices, e.g., objective clipping, KL penalties, relative rewards, and so on. Compared with these baselines, our method actually simplifies and removes many sophisticated designs (including objective clipping, relative rewards, etc.).
>
> We hope the above response resolves your concerns and helps you re-evaluate our work. Please do not hesitate to let us know if you have any further questions or require additional clarification.

---

### Official Review · Reviewer_bdbs · 2025-11-01

**Soundness:** 2
**Presentation:** 2
**Contribution:** 2
**Rating:** 2
**Confidence:** 3

**Summary:**

The authors propose a new RL method, Confidence Calibrated Group Sequence Policy Gradient (C2SPG), that aims to both improve accuracy and calibrated model confidence on reasoning tasks. The authors define confidence as the average token-level likelihood of the output under the language model. The proposed RL algorithm introduces several changes to the optimization target from prior work, such as normalizing the advantage estimate by 1 - confidence of the old model (used to generate trajectories), multiplying the advantage estimate by the log of the confidence, and introducing a "binary cross entropy regularizer" that is the log-likelihood that the output is correct when treating the average token likelihood as the predicted probability. The authors analyze the gradients of the proposed algorithm and compare to prior methods. They also consider extensions to non-binary settings.

The authors perform experiments on a dataset with logic puzzles (K&K) and across multiple mathematical reasoning datasets. They report dramatic improvements in accuracy (38% GRPO, 56% GSPO --> 79% C2SPG) and calibration (0.5 GRPO, 0.3 GSPO --> 0.13 ECE) on the K&K dataset, and mixed performance on mathematical reasoning tasks (depending on hyperparameters of the C2SPG objective, it achieves up to 1% increase in average accuracy with worse calibration or ~0.03 improvement in ECE to 0.46 with worse accuracy).


The authors claim that C2SPG "significantly improves both reasoning accuracy and confidence calibration, outperforming existing state-of-the-art methods".

**Strengths:**

The authors perform an interesting comparison of RL methods and use this to motivate their work. Experiments are clearly defined and presented.

**Weaknesses:**

1. The current experimental results do not seem to support the claim that C2SPG significantly improves calibration and reasoning. On the math benchmarks, C2SPG either has a small improvement in accuracy while achieving similar calibration or improves calibration while achieving similar accuracy. These results don't tell a clear story of improvement or show a single algorithmic setting that improves both of the target metrics. Additionally, small scale of experiments (100 steps, 3B models) and lack of large improvement on math benchmarks makes it unclear if the improvements that are seen here will generalize to other experimental settings or scale up.

2. The results from the two experiments are not consistent: the accuracy improvement on K&K relative to SOTA is dramatic (38% GRPO, 56% GSPO --> 79% C2SPG), while the accuracy improvement on math is marginal (~1-2% on best hyper parameter setting). Why is the improvement so big on the first experiment and not on the second? As is, it is not clear how informative the K&K setting is, given that it is synthetic and behaves so differently from the math benchmarks.

3. It's not clear to me that the average token-level likelihood of a sequence should correspond to model confidence, or that it is desirable to optimize this quantity for calibration. There is a mismatch between this quantity and the sequence likelihood (which depends on length), so a sequence that the model is more likely to generate may be defined to have lower confidence. It's not clear what the implications of this mismatch are and why it is desirable. I note that this settings is different from the GSPO setting discussed in the paper: GSPO length normalizes sequence likelihoods when doing importance weighting, which means that sequences being compared will always be the same length. Stronger justification of why this definition of confidence is the right one for developing calibrated models, and why this is a robust choice, would be helpful.

4. The method proposes a number of changes to previous RL methods (see summary) without clearly motivating all of them and ablating these choices. Some other simple approaches, such as treating the confidence as the predicted probability and simply optimizing the log-likelihood that the output is correct given this probability, are not compared against. It is a bit difficult to understand what choices could be yielding improvements and why some choices are made without this kind of discussion and analysis.

**Questions:**

Please see the weaknesses section.

---

> ### Author Response · Authors · 2025-11-20
> **Response to Reviewer bdbs (Part 1)**
>
> Below, we answer your questions one by one. All required experiments are included in our revised paper, and some are shown below for quick review.
>
> **Q1: Test on larger LLMs**
>
> **A1:** Following AR-Lopti, we fine-tune Qwen2.5-7B for math reasoning and Qwen2.5-7B-Instruct-1M for logic reasoning. **Tables 7 and 8 in Appendix D.1 show that our method outperforms baselines consistently on larger LLMs, demonstrating its scalability.**
>
> ||GSM8K|MATH500|AMC23|AIME24|Olympiad|Minerva|Acc|BS|ECE|
> |-|-|-|-|-|-|-|-|-|-|
> |GRPO|89.60|72.04|43.87|16.33|**25.10**|23.82|45.13|0.531|0.535|
> |GPG|89.63|**73.24**|51.56|13.00|24.62|23.75|45.97|0.515|0.520|
> |GSPO|88.75|72.44|**53.19**|14.67|24.56|22.35|45.99|0.520|0.524|
> |AR-Lopti|88.72|73.08|47.31|14.42|24.65|**24.26**|45.41|0.522|0.528|
> |C²GSPG|**90.60**|71.56|52.12|**20.67**|24.44|22.94|**47.06**|**0.511**|**0.515**|
>
> ||3|4|5|6|7|Acc|BS|ECE|
> |-|-|-|-|-|-|-|-|-|
> |GRPO|79.00|72.40|56.40|56.60|42.40|61.36|0.254|0.308|
> |GPG|90.60|88.20|73.20|69.60|53.60|75.04|0.190|0.194|
> |GSPO|84.00|88.00|75.20|72.00|67.20|77.28|0.157|0.171|
> |AR-Lopti|82.60|87.60|71.60|68.00|58.40|73.64|0.180|0.214|
> |C²GSPG|**95.60**|**92.80**|**85.80**|**77.00**|**73.00**|**84.84**|**0.126**|**0.124**|
>
> **Q2: Analyze the results**
>
> **A2:** Our method performs well on K&K but incrementally in math reasoning. **This phenomenon is common [5-7] due to the nature of these two datasets.**
>
> 1) K&K is an unseen but simple task. It is based on the Knights and Knaves reasoning setting [1], with rule-based rewards [2]. K&K does not appear in the pre-training corpora of mainstream LLMs, making it an excellent dataset for evaluating RL methods [3-7] -- performance improvements primarily stem from learning rather than data memorization.
>
> 2) MATH is a seen but difficult task. Although LLMs were trained on math content, their capabilities are still insufficient to solve MATH problems. Current RL methods, including ours, cannot exceed the upper bound of these LLMs' capabilities.
>
> [1] On memorization of large language models in logical reasoning, 2024
>
> [2] Logic-rl: Unleashing LLM reasoning with rule-based reinforcement learning, 2025
>
> [3] REINFORCE++: Stabilizing Critic-Free Policy Optimization with Global Normalization, 2025
>
> [4] DUMP: Automated Distribution-Level Curriculum Learning for RL-based LLM Post-training, 2025
>
> [5] A Simple "Motivation" Can Enhance Reinforcement Finetuning of Large Reasoning Models, 2025
>
> [6] Do Not Let Low-Probability Tokens Over-Dominate in RL for LLMs, 2025
>
> [7] Efficient RL Training for Reasoning Models via Length-Aware Optimization, 2025
>
> **Q3: The rationality of the confidence definition using the average token-level likelihood**
>
> **A3:** We use the average token-level likelihood of a sequence as the model confidence for two reasons.
>
> 1. **The rationale and origin of our definition.** The confidence we defined constitutes a standard white-box estimation relying on the LLM's internal states [1,2]. Prior studies [2–4] utilize similar token-level likelihood to quantify confidence. Essentially, this definition shares the same underlying principle as perplexity [5–9], and these two metrics are inversely related. Consequently, our definition is reasonable and follows long-standing protocols already validated in prior work [2-9].
>
> 2. **Normalization in our definition resolves the sequence length mismatch problem.** Sequence likelihood depends on length. That’s why we shouldn't use it as the definition of confidence. Taking sequence likelihood, rather than the average token-level likelihood, as confidence will introduce overly strong bias on long sequences and penalize the contribution of short sequences during training (see our objective, in which log(confidence) is used as loss), which produces the mismatch you mentioned. In fact, both our method and GSPO apply length normalization so that sequences being compared are always of the same length. This setting is useful for RL reasoning, ensuring equitable comparisons across generated sequences of varying lengths.
>
> [1] Shifting attention to relevance: Towards the predictive uncertainty quantification of free-form large language models, 2024
>
> [2] Contextualized sequence likelihood: Enhanced confidence scores for natural language generation, 2024
>
> [3] Language model cascades: Token-level uncertainty and beyond, 2024
>
> [4] Benchmarking uncertainty quantification methods for large language models with lm-polygraph, 2024
>
> [5] Evaluation metrics for language models, 2023
>
> [6] Fact-Checking the Output of Large Language Models via Token-Level Uncertainty Quantification, 2024
>
> [7] Mitigating Forgetting in LLM Fine-Tuning via Low-Perplexity Token Learning, 2025
>
> [8] What is Wrong with Perplexity for Long-context Language Modeling?, 2024
>
> [9] Can Perplexity Predict Fine-tuning Performance? An Investigation of Tokenization Effects on Sequential Language Models for Nepali, 2024

---

> ### Author Response · Authors · 2025-11-20
> **Response to Reviewer bdbs (Part 2)**
>
> **Q4: Add more ablation studies**
>
> **A4:** The motivations of the modules in our method have been explained in Lines 192–204. We provide explanations below for quick review.
>
> 1. **Normalizing the advantage estimate by 1-confidence of the old model:** It introduces model confidence as a fine-grained modulation factor across sequences within the same group and encourages the model to be more confident for truly advantageous sequences while reducing confidence for disadvantageous ones.
>
> 2. **Multiplying the advantage estimate by the log of the confidence:** It pursues the consistency between confidence and the advantage function, i.e., a high-confidence sequence should receive a high advantage.
>
> 3. **BCE regularization:** It penalizes the discrepancy between the reward and the model confidence of the sequence. This regularizer pushes the model confidence toward 1 for high-reward sequences and toward 0 for low-reward sequences.
>
> To further address your concern, we designed an ablation study on K\&K using Qwen2.5-3B-Instruct. For each method, we augment it with our BCE regularizer. **Table 9 in Appendix D.2 shows that a) using the confidence-based advantage, GSPG outperforms the baselines, and b) the BCE regularizer helps improve all the methods. These results demonstrate the usefulness of different modules.**
>
> ||3|4|5|6|7|Acc|BS|ECE|
> |-|-|-|-|-|-|-|-|-|
> |GRPO|58.67|44.93|34.53|31.87|20.53|38.11|0.405|0.497|
> |GRPO+BCE|66.60|57.07|45.80|41.13|29.73|48.07|0.335|0.409|
> |GPG|69.53|64.80|49.47|43.20|35.93|52.59|0.363|0.363|
> |GPG+BCE|70.93|66.07|53.07|43.87|36.33|54.05|0.337|0.330|
> |GSPO|71.13|66.40|55.40|48.20|40.07|56.24|0.291|0.302|
> |GSPO+BCE|77.07|76.53|61.80|55.73|47.27|63.68|0.231|0.233|
> |AR-Lopti|73.20|64.93|54.13|49.47|41.33|56.61|0.282|0.343|
> |AR-Lopti+BCE|78.40|81.80|70.27|66.13|58.93|71.11|0.192|0.225|
> |GSPG|79.50|75.50|67.00|57.00|50.00|65.80|0.215|0.256|
> |GSPG+BCE(our C²GSPG)|**86.00**|**88.00**|**78.60**|**73.27**|**69.00**|**78.97**|**0.136**|**0.126**|
>
> **Q5: Provide simple baselines like SFT**
>
> **A5:** If our understanding is correct, “simply optimizing the log-likelihood that the output is correct” is equivalent to standard SFT (i.e., optimizing the probability of correct tokens). To evaluate this, we train an SFT baseline on the K&K dataset. Table 13 in Appendix D.6 shows that SFT performs worse than RL methods across all accuracy and calibration metrics, demonstrating that merely optimizing the log-likelihood of correct answers is insufficient.
>
> ||3|4|5|6|7|Acc|BS|ECE|
> |-|-|-|-|-|-|-|-|-|
> |Base|12.00|13.00|2.00|2.00|0.00|5.80|0.554|0.705|
> |SFT|38.00|33.00|18.00|18.00|8.00|23.00|0.506|0.623|
> |GRPO|58.67|44.93|34.53|31.87|20.53|38.11|0.405|0.497|
>
> We hope our response resolves your concerns and helps you re-evaluate our work. Please do not hesitate to let us know if you have any further questions or require additional clarification.

---

### Author Response · Authors · 2025-11-26
**Gentle Reminder**

Dear Reviewers,

We sincerely appreciate the time and effort you devoted to providing thoughtful and valuable feedback. We have carefully addressed all your concerns in our point-by-point responses and have uploaded a revised paper, with new content highlighted in blue.

As the author-reviewer discussion phase is approaching its end, we would be grateful if you could take a moment to review our responses and revisions. If you have any remaining questions or concerns, we are more than happy to engage in further discussion and provide additional clarification.

Best regards,

Authors of ICLR 2026 Submission 10828

---

### Author Response · Authors · 2025-12-01
**Summary of the Rebuttal**

Dear Reviewers, ACs, SACs, and PCs,

In the past few days, we have done our best to address reviewers' concerns and revise our submission, including adding suggested analytical experiments, explaining the rationale for the proposed method, and providing more detailed analysis of model performance.

Unfortunately, **we submitted our rebuttals on Nov. 21 but have not received any feedback until the security incident happened.** Now, we are no longer able to interact with the reviewers and have no way to know whether they are satisfied with our response.

Because our work received mixed scores across a wide range (**8**, **6**, **4**, and **2**), we would like to summarize our rebuttal for your convenience in the following AC-reviewer discussion and decision phases.

**Main Concerns of Reviewers**

-**Reviewer bdbs (score 2)**

1) Test on larger LLMs.

2) On MATH datasets, reasoning accuracy and confidence calibration are in conflict, and the improvement is incremental.

3) The rationality of the confidence we defined.

4) Add ablation studies on the modules of the proposed method.

-**Reviewer CHKD (score 4):**

1) Why is overconfidence an important question?

2) Will calibrating confidence lead to performance degradation?

3) Add ablation studies on the modules of the proposed method.

-**Reviewer QD6C (score 6):**

1) Confidence calibration v.s. reward calibration.

2) Test on larger LLMs.

3) The improvements on MATH datasets are incremental.

-**Reviewer s7fz (score 8):**

1) Add ablation studies on the modules of the proposed method.

2) Test on larger LLMs.

3) Apply to test-time scaling.

4) Test robustness to hyperparameters.

**Summary of Our Response**

During the rebuttal phase, we addressed all the reviewers' questions one by one and supplemented our work with additional experiments and explanations. Key updates include:

- We add experiments on 7B models and consider more baselines (**Tables 7, 8, 13**), demonstrating that **our method can scale up to larger LLMs and outperform baselines consistently**. Moreover, **Table 12** shows that our method is **applicable to test-time scaling**.
- We add ablation studies on the modules of our method (**Table 9**), including the GSPG framework and the BCE regularizer, demonstrating that **merely our GSPG has already outperformed baselines, and the BCE regularizer further helps improve performance.**
- We add ablation studies on the hyperparameters $\beta$ and $\alpha$ (**Tables 10, 11, and Figure 7**), demonstrating **the rationality of our settings** and the robustness of our method to the hyperparameters.
- We demonstrate that with the change of $\beta$, **the improvements of our method on reasoning accuracy and confidence calibration are consistent** on the K&K dataset (**Table 10**) but conflicted on the MATH datasets. We explained this phenomenon in terms of model capability and task difficulty in **Section 4.2**.
- We add explanations and references, supporting the definition of our confidence.

**In summary, although no one has provided feedback, we carefully addressed all points raised in the initial review. We believe these concerns have been resolved and no longer constitute grounds for rejection.** In light of this, we respectfully hope that our paper, the reviews, and our detailed rebuttals can be considered and discussed sufficiently in the following phases. Thanks in advance for your fair and careful assessment.

Best regards,

Authors of Submission 10828

---

### Meta-Review · Area_Chair_iuJL · 2026-01-07

**Summary:**

Reviewer bdbs questioned the measure of confidence that you use. While I of course agree that length-normalized probability has some advantages over unnormalized probability in these settings, it also has some significant disadvantages. For example, consider a case where the task is to generate strings like "Here are 12 xs: xxxxxxxxxxxx." while varying the number 12. A perfect language model in this task would be absolutely certain on every step except for the tokens corresponding to the number, and the unnormalized probability would exactly align with the probability of generating any given number $n$. Assuming a one-token parameterization of numbers and of each $x$, your measure of confidence would be $p(n) / (n + c) + (1 - 1/(n+c))$, and so strings with larger $n$ are treated as substantially more "confident" than they actually are. This makes your Brier score/ECE/etc fundamentally incorrect: a model that is actually calibrated (matches the desired $p(n)$) would be treated as worse than one that substantially underestimates the probability of larger $n$. This is of course a contrived setting, but it is not altogether unreasonable in relation to mathematical settings, which might have a "hinge point" where a mistake is made but then long nearly-certain calculations following on from there. Note that this is not addressed by the separation of the problem format into `<think>` and `<answer>` sections, since the `<answer>` section will typically be highly-confident given the `<think>`.

More broadly, I'm not sure that traditional notions of calibration are even really what we "want" LLMs to satisfy. If the answer is uncertain and we generate many responses, we get a variety of answers. But practical uses of LLMs don't typically generate many responses, and if they do, they're usually taking the mode or similar anyway. What a user interaction with a language model would "want" is something that says "I don't really have enough information to tell here, but it's probably X or it could be Y, for these reasons." On a question where the answer is a 50-50 coin flip, a model that says "50% X, 50% Y" with perfect certainty would be treated as being horribly overconfident with your scheme, while a model that alternatively says "It's X", "The answer is X", ... could be treated as perfectly calibrated (if the length-normalized uncertainties work out appropriately). Calibration in these settings is really about the abstract notion of an "answer," _not_ about the tokenized string being produced by the language model. The mapping from strings to the "answer" is complex, and so handling this fully is of course difficult. But I think it is somewhat misguided to not even attempt to address this question, and can easily mislead in the quest towards better meaningfully calibrated models. (The papers you cite of Yoon et al. and Damani et al., as well as Kadavath et al., take steps in this direction, but you only cite them without any further discussion.)

----

The reviewers had other concerns which I think were partially addressed by the rebuttal. For example, the addition of ablation studies was helpful and appreciated. I share some of these remaining concerns, however; for example, only testing on the Qwen family is somewhat concerning for a supposedly-general approach. Having at least a few experiments tuning a Llama, Pythia, or other open-source models – even ones that are not necessarily larger, but at least "different" – would be a significant advantage to the empirical results here. Given the modest improvements and lack of theoretical grounding, as well as foundational questions about the applicability of the approach, I think this submission falls somewhat short of the high bar for ICLR this time around. I encourage the authors to consider a little more the foundational issues around studying calibration in this space (at least with some discussion and investigation into how the length of reasoning chains connects with your confidence scores), add some experiments on at least one other model class, and address the other remaining reviewer concerns before resubmitting to a future venue.

**Reviewer Concerns:**

Discussed above.

**Reviewer Scores:**

Hard to know; I think some of the issues were adequately addressed, but others were not, so I do not imagine that scores would have drastically increased.

---

### Decision · Program_Chairs · 2026-01-26

Reject